# Conformational dynamics of the Beta and Kappa SARS-CoV-2 spike proteins and their complexes with ACE2 receptor revealed by cryo-EM

Yifan Wang [1,2,4], Cong Xu[1,4], Yanxing Wang[1,4], Qin Hong[1,2,4], Chao Zhang [3,4], Zuyang Li[1,2], Shiqi Xu [3], Qinyu Zuo[1], Caixuan Liu[1,2], Zhong Huang [3✉] & Yao Cong [1,2✉]

The emergence of SARS-CoV-2 Kappa and Beta variants with enhanced transmissibility and resistance to neutralizing antibodies has created new challenges for the control of the ongoing COVID-19 pandemic. Understanding the structural nature of Kappa and Beta spike (S) proteins and their association with ACE2 is of significant importance. Here we present two cryo-EM structures for each of the Kappa and Beta spikes in the open and open-prone transition states. Compared with wild-type (WT) or G614 spikes, the two variant spikes appear more untwisted/open especially for Beta, and display a considerable population shift towards the open state as well as more pronounced conformational dynamics. Moreover, we capture four conformational states of the S-trimer/ACE2 complex for each of the two variants, revealing an enlarged conformational landscape for the Kappa and Beta S-ACE2 complexes and pronounced population shift towards the three RBDs up conformation. These results implicate that the mutations in Kappa and Beta may modify the kinetics of receptor binding and viral fusion to improve virus fitness. Combined with biochemical analysis, our structural study shows that the two variants are enabled to efficiently interact with ACE2 receptor despite their sensitive ACE2 binding surface is modified to escape recognition by some potent neutralizing MAbs. Our findings shed new light on the pathogenicity and immune evasion mechanism of the Beta and Kappa variants.

[1] State Key Laboratory of Molecular Biology, National Center for Protein Science Shanghai, Shanghai Institute of Biochemistry and Cell Biology, Center for Excellence in Molecular Cell Science, Chinese Academy of Sciences, 200031 Shanghai, China. [2] University of Chinese Academy of Sciences, 100049 Beijing, China. [3] CAS Key Laboratory of Molecular Virology and Immunology, Institut Pasteur of Shanghai, Chinese Academy of Sciences, University of Chinese Academy of Sciences, 200031 Shanghai, China. [4] These authors contributed equally: Yifan Wang, Cong Xu, Yanxing Wang, Qin Hong, Chao Zhang. ✉email: huangzhong@ips.ac.cn; cong@sibcb.ac.cn

Severe acute respiratory syndrome coronavirus 2 (SARS-CoV-2) is an infectious agent responsible for the ongoing coronavirus disease 2019 (COVID-19) pandemic. The spike (S) glycoprotein of SARS-CoV-2 mediates receptor recognition and viral entry into cells[1–5]. It forms homotrimers protruding from the virus surface and, once engaged with the host-cell receptor–human ACE2, undergoes a substantial structural rearrangement to fuse the viral membrane with the host-cell membrane[1,3–11]. The S protein is also the primary target of the humoral immune response during infection. A large number of SARS-CoV-2 neutralizing monoclonal antibodies bind S protein, especially its receptor-binding domain (RBD), providing a basis for vaccine development[12–24]. Clearly, the SARS-CoV-2 S protein plays a critical role in the spread and tropism of the virus as well as its ability to provoke and evade the immune system[25].

SARS-CoV-2 undergone considerable evolution since its initial discovery in late 2019. A number of SARS-CoV-2 lineages were defined as variants of concerns (VOCs) by the World Health Organization (WHO), including the B.1.1.7 (Alpha) lineage that arose in the UK[26–30], B.1.351 (Beta) lineage in South Africa[29–33], P.1 (Gamma) lineage in Brazil[34], and B.1.617.2 (Delta) lineage in India[35,36]; while the B.1.617.1 (Kappa) variant was defined as a variant of interest (VOI)[37–41]. These variants carry multiple mutations in S protein and some of them show enhanced transmissibility and resistance to antibody neutralization[25]. In particular, the Beta S protein contains the D614G substitution and additional nine mutations, including a cluster of mutations in the N-terminal domain (NTD), three substitutions (K417N, E484K, and N501Y) in the receptor-binding domain (RBD), and one substitution (A701V) near the furin cleavage site. The Kappa variant also harbors multiple mutations in the S protein. Thus far, there are structural studies focused on the free S protein of Beta variant[29,30], yet no available structures on the Kappa-S trimer and S in complex with ACE2 receptor for both Kappa and Beta variants.

Here, we present cryo-EM structures of the S trimer of the SARS-CoV-2 Kappa and Beta variants in the open or transition state for each variant at the resolution of 3.2–3.6 Å, revealing not only their unique conformational dynamics potentially related to enhanced virus fitness, but also altered antigenic surfaces permitting immune escape. Moreover, we captured four conformational states for both Kappa and the Beta S trimers engaged with the human ACE2 receptor at 3.6–4.0-Å resolution. Combined with 3D variability analysis (3DVA), we depicted their enlarged conformational landscape and population distribution shift relative to the WT S-ACE2 complex[8], and continuous conformational transitions between different states. Our biochemical and structural analyses of the interaction between RBD and ACE2 showed that the two variant S proteins can efficiently recognize and bind ACE2 despite that the sensitive ACE2-binding surface is modified to escape antibody recognition.

365, 84, and 83 nM, respectively (Fig. 1a). Overall, the G614 S bound ACE2 less tightly than did the WT S, in general consistent with most of the related reports[42,43], despite seemingly difference from a previous report[44], likely caused by the ACE2 protein used (monomer vs dimer) and the biosensor loading method (with ACE2 or S). The Beta and Kappa S proteins, both of which bear D614G mutation, showed stronger ACE2 binding than did the G614 S, suggesting that the other mutations within the Beta and Kappa S may enhance receptor recognition. The S/ACE2 affinity for the two variants is also slightly stronger than that of WT S. Overall, our BLI data are consistent with the S/ACE2 affinity results for Beta/Kappa variants reported recently[29,30,45,46].

We then compared the WT, Beta, and Kappa S trimers for their binding with two well-characterized neutralizing MAbs, 2H2 and 3C1[23]. As shown in Fig. 1b, all three S trimers efficiently reacted with 3C1, which mainly targets the core region of RBD, in an antigen dose-dependent manner despite slightly decreased reactivity was observed for the Beta S trimer. In contrast, both the Beta and Kappa S trimers failed to react with 2H2, which mainly binds the RBM region of the WT RBD, regardless of the antigen doses (Fig. 1b). These S trimer-binding data were consistent with the results from SARS-CoV-2 pseudovirus-neutralization assays, which showed that the Beta and Kappa pseudoviruses remained sensitive to 3C1 but were refractory to neutralization by 2H2 (Fig. 1c). Specifically, the calculated $IC_{50}$ values of 2H2 against the Beta or Kappa pseudoviruses were both determined to be >10 μg/ml (Fig. 1c), in sharp contrast to that (25 ng/ml) against the WT pseudovirus[23].

The Beta SARS-CoV-2 variant carries multiple mutations on the S protein, three of which are located in the RBD region, including K417N, E484K, and N501Y[31,32]. MAbs 2H2 and 3C1 mainly target the RBM and core region of the RBD of the original WT S (Fig. 1e)[23]. Therefore, to evaluate the impact of K417N, E484K, and N501Y mutations on 2H2 or 3C1 binding, we generated a panel of mutant RBD proteins carrying single or triple mutations, including RBD (K417N), RBD (E484K), RBD (N501Y), and Beta-RBD (carrying the K417N-E484K-N501Y triple mutations). The four RBD mutants were compared in parallel with the WT RBD for reactivity with 2H2 or 3C1 in ELISA (Fig. 1d). The result showed that MAb 2H2 retained reactivity with RBD (K417N) and RBD (N501Y) but failed to bind RBD (E484K) and the triple mutant (Beta-RBD), indicating that E484K is responsible mainly for the loss of 2H2 binding. In contrast, 3C1 showed comparable binding activity with all four mutant RBDs as well as the WT one as did the polyclonal anti-RBD sera, suggesting that the three mutation sites are not involved in 3C1 binding, in agreement with the structural findings from our previous study[23].

Taken together, the above data show that, compared with the WT S, the Beta and Kappa-S proteins possess distinct binding profiles to ACE2 receptor and neutralizing antibodies.

## Results

**Binding properties of the Kappa and Beta S proteins to ACE2 receptor and neutralizing antibodies**. A panel of trimeric S proteins representing an original SARS-CoV-2 strain (Wuhan-Hu-1, hereafter referred to as wild type, WT), an early-phase variant with D614G mutation (hereafter referred to as G614), Beta, and Kappa variants, respectively, were generated and tested for their binding to ACE2 receptor by biolayer interferometry (BLI) assay (Fig. 1a and Supplementary Fig. 1). In the current study, to be consistent with our following structural investigation, we used monomeric ACE2 (human ACE2 PD domain) and stabilized S ectodomain for BLI analyses. Our BLI results showed that the $K_D$ values for WT, G614, Kappa, and Beta S were 104,

**Architecture and conformational dynamics of the Kappa variant S trimer**. We carried out cryo-EM study to delineate structural changes of the Kappa S trimer, and obtained two cryo-EM maps, including a one RBD-up open conformation (termed Kappa S-open) and a transition state (termed Kappa S-transition) at 3.2- and 3.4-Å resolution, respectively (Fig. 2a, c and Supplementary Fig. 2a–d and Supplementary Table 1). We then built a model for each of the structures (Fig. 2b and Supplementary Fig. 2e). The protomer with the "up" RBD is referred to as protomer 1. For the open state, the angle between the long axis of the up RBD-1 and the horizontal plane of S trimer is about 72.5° (Fig. 2d); while for the transition state, this angle is 44.1° with the

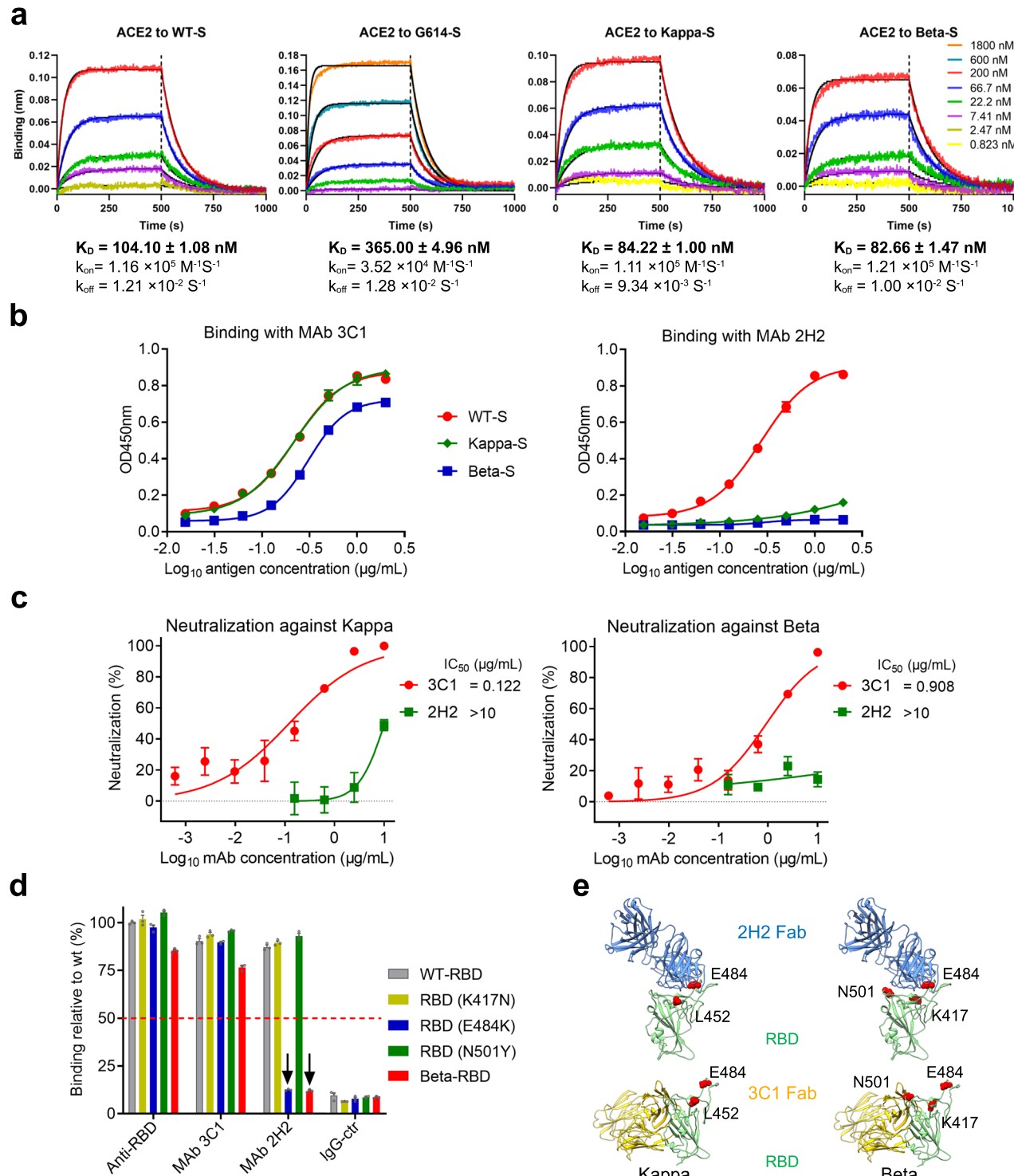

**Fig. 1 Characterization of properties of the S proteins of Beta and Kappa variants. a** Measurement of binding affinity between S trimers and ACE2 using biolayer interferometry (BLI). Association and dissociation steps are divided by dotted lines. Raw sensorgram curves and fitting curves were shown in color and black, respectively. **b** Reactivity of the WT, Beta, and Kappa S trimer proteins with the MAbs 3C1 and 2H2 were determined by ELISA. Data are expressed as mean ± SD of triplicate wells. **c** Neutralization activity of MAbs 3C1 and 2H2 against Kappa and Beta pseudoviruses. Data are expressed as mean ± SEM of four replicate wells. **d** Binding of the antibodies to WT and mutant RBD proteins were measured by ELISA. Anti-RBD polyclonal antibody served as a positive control. Binding level of anti-RBD polyclonal antibody to WT RBD was set to 100%. The red dotted line represents cutoff value (50%). Downward arrows indicate that binding signals of the corresponding mutants greatly reduced as compared to WT RBD. Data are mean ± SEM of triplicate wells. **e** Illustration of the distinct binding sites of MAbs 2H2 (PDB: 7DK4, cornflower blue) and 3C1 (PDB: 7DCC, gold) on the RBD (light green) of the original WT S. The mutation sites (mapped on the WT RBD as a red sphere) of the two variants are all located in the binding footprint of 2H2 on the WT RBD.

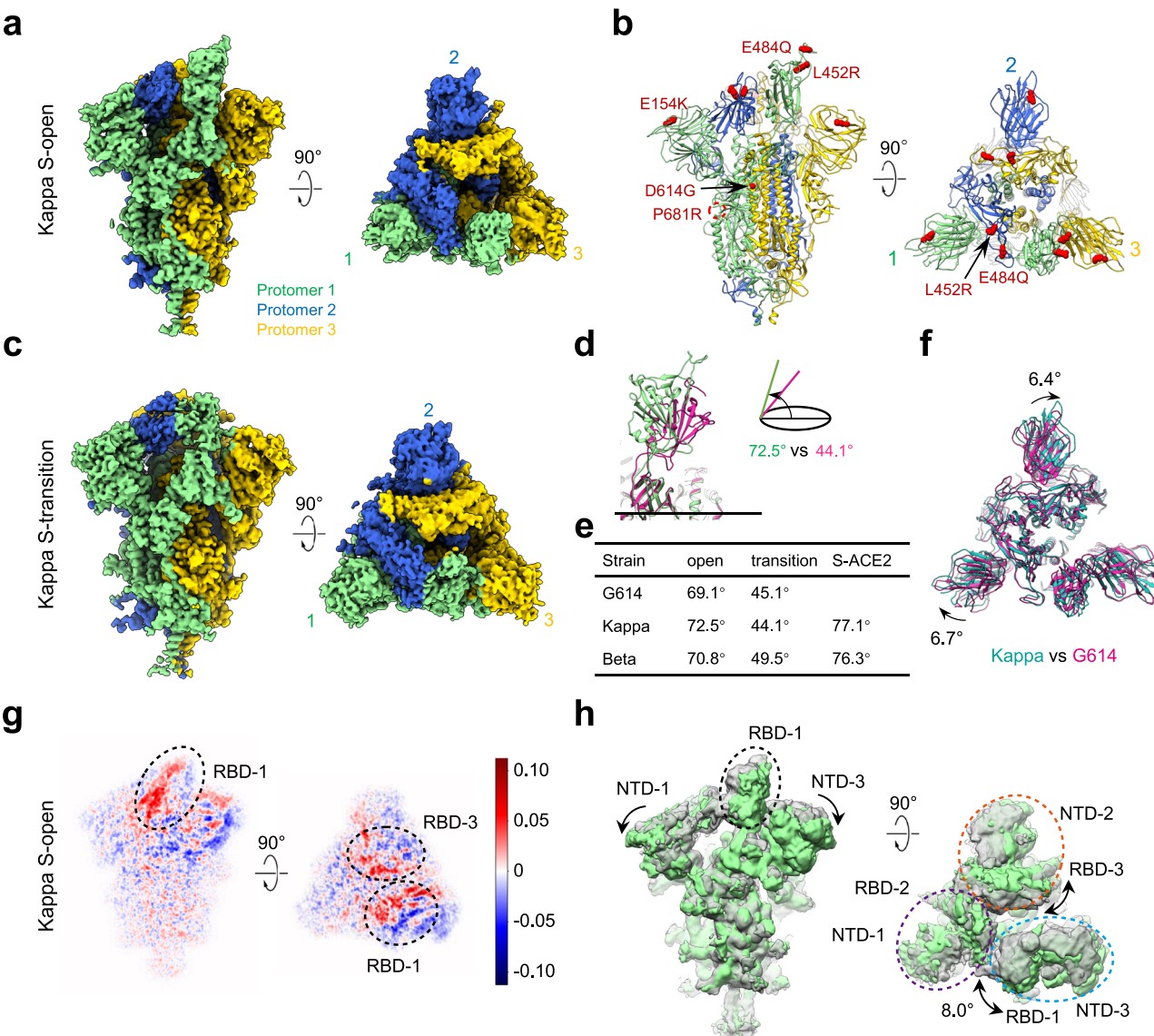

**Fig. 2 Cryo-EM structures of Kappa variant S trimer. a, b** Cryo-EM map (**a**) and model (**b**) of the Kappa S-open, with protomer 1, 2, and 3 shown in light green, royal blue and gold, respectively. This color scheme is followed throughout. The mutations of Kappa S are shown as red sphere in the model. **c** Cryo-EM map of the Kappa S-transition. **d** Side view of the overlaid RBD-1 from S-open (light green) and S-transition (violet red), showing that the angle between the long axis of RBD and the horizontal plane of S trimer reduces from S-open to S-transition. **e** The angle between the long axis of RBD-1 and the horizontal plane of S trimer in different variants. **f** Top view of the overlaid structures between the open state of Kappa (in light sea green) and G614 (PDB: 7KRR, violet red). **g, h** A representative 3DVA motion of the Kappa S-open dataset, displayed as central slices in which positive (red) and negative (blue) values correspond to density to be added and subtracted from the mean density (**g**), and in maps (**h**) showing two extreme conditions in the variance, with the angular range, direction of the motion, and the three NTD-RBD pairs displayed. The rotation axis for RBD is around the lower part of SD1 (also see Supplementary Fig. 5c). This motion rendering style is followed throughout.

RBD-1 appearing less "up", and the other portion of S-transition is in similar conformation to that of the S-open (Fig. 2d, e and Supplementary Fig. 2f). Moreover, the population distribution of the S-open versus S transition is ~49.1–50.9% (Supplementary Fig. 2b), in sharp contrast to our previous findings that, for the WT S trimer, the S-open occupies only 6% and the S-closed about 94% of the population[8]. These data indicate a much more fusion-prone status for the Kappa S trimer. Moreover, compared with the open state of a parental strain G614 S trimer (PDB ID: 7KRR)[43], the Kappa S-open appears more untwisted/open, with the NTD-1 and NTD-2 clockwise rotated 6.7° and 6.4°, respectively (Fig. 2f and Supplementary Fig. 2g), and the NTDs outward/downward tilted slightly, leading to a reduced protomer interaction. Accordingly, with the downward movement of NTD

and the underneath SD1, the fusion peptides (FPs) of the Kappa S-open were also disordered as in the other open state S trimer structures[8,47–49]. In the meanwhile, the up RBD-1 slightly upward tilted about 3.4° (from 69.1° to 72.5°) relative to that of the G614 S (Fig. 2e).

To capture the continuous conformational dynamics of the Kappa S trimer, we performed further 3DVA on the Kappa S-open dataset through cryoSPARC[50]. Interestingly, the analysis revealed a "breath" motion of the fusion machinery, especially in the S1 region (Supplementary Movie 1). In the collective motions, the three coordinated NTD-RBD pairs (including NTD-1/RBD-2, NTD-2/RBD-3, and NTD-3/RBD-1) tilt outward/downward simultaneously, thus the entire machine untwists and appears more expanded (Supplementary Movie 1 and Fig. 2g, h), which

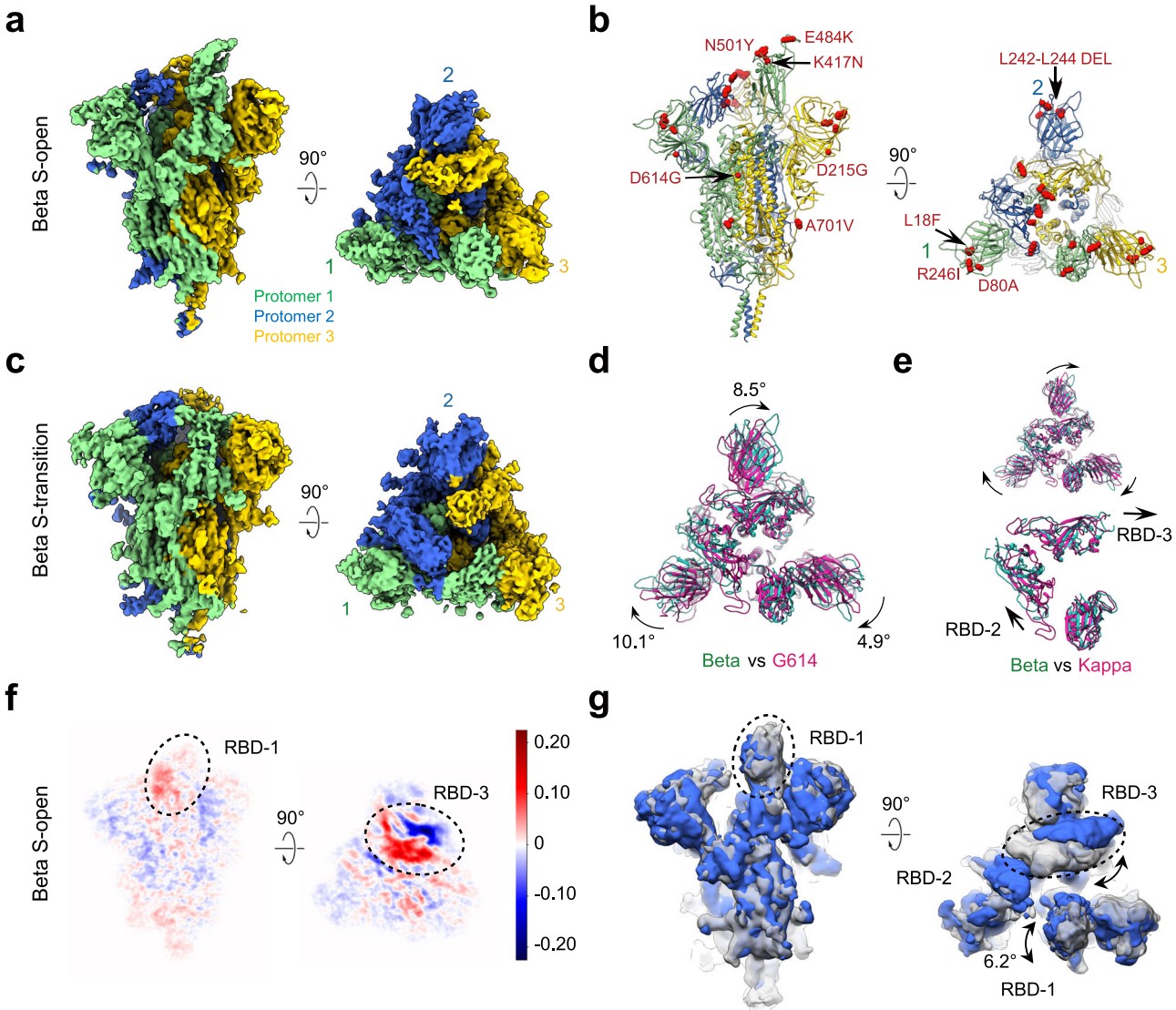

**Fig. 3 Cryo-EM structures of Beta S variant trimer. a**, **b** Cryo-EM map (**a**) and model (**b**) of Beta S-open. The mutations of Beta S are shown as red sphere in the model. **c** Cryo-EM map of Beta S-transition. **d** Structural comparison between Beta S-open (in light sea green) and G614 S-open (PDB: 7KRR, violet red). **e** Structural comparison between Beta S-open (in light sea green) and Kappa S-open (in violet red), and zoom-in view of RBDs. **f**, **g** A representative 3DVA motion of the Beta S-open dataset, displayed as central slices (**f**), and in maps (**g**) showing two extreme conditions in the variance.

could release the protomer interaction strength, beneficial for the transient raising up of the RBD and shedding of S1 subunits. We showed previously that for the WT S, the core region of the up RBD-1 (through Y369/F374) can form aromatic interactions with the RBM T470-F490 loop of the neighboring down RBD-2 (through F486/Y489)[8]. Here for the Kappa variant S with E484Q mutation in the T470-F490 loop, contacts between RBD-1 and RBD-2 within 4 Å were reduced to just between Y369 of RBD-1 and N487/A475 of RBD-2, which could disturb the constrains between RBD-1 and RBD-2, facilitating conformational landscape shift towards the fusion-prone open state. All these may render the Kappa variant S more prone to receptor binding and subsequent fusion.

**Architecture and conformational dynamics of the Beta variant S trimer.** We also resolved two cryo-EM maps of the Beta S trimer, including a one RBD-up open state (termed Beta S-open) and a transition state (termed Beta S-transition) at 3.5- and 3.6-Å resolution, respectively (Fig. 3a, c, Supplementary Fig. 3a–d, and Supplementary Table 2). We then built a model for each of the

structures (Fig. 3b and Supplementary Fig. 3e). The proportions of the Beta S-open and S-transition are ~53.1% and 46.9%, respectively (Supplementary Fig. 3b). For the Beta S-open, the angle between the up RBD-1 and the horizontal plane of S trimer is about 70.8°, slightly smaller than that of the Kappa S-open (Fig. 2e). For the transition state, the RBD-1 appears less "up" (the RBD-1 tilting angle is about 49.5°), and the RBD-1/2 appear more dynamic than that in S-open with density for large parts of RBM missing. In the meanwhile, the other portion of the S transition is in similar conformation to S-open (Supplementary Fig. 3f). Noteworthy, for Beta S-open, the NTD-1/2 exhibit clockwise rotation of 10.1° and 8.5° relative to that of the G614 open state (Fig. 3d), making the machinery appear even more untwisted and less compact than that of the Kappa S-open (Fig. 3e). In addition, its RBD-2/3 shift outward relative to the counterparts of Kappa S-open (Fig. 3e), resulting in loss of contacts between RBD-1/2, which, together with larger gaps between RBD-2/3, makes the trio of RBDs appear to lose constrains and be more separated as compared to that of the S-open of Kappa variant.

Our 3DVA analysis on the Beta S-open dataset revealed a motion showing that with the down/up movement, RBD-3 (belonging to S1 subunit) contacts/leaves the S2 HR1-CH hairpin from the neighboring protomer 2, which could be propagated to the S2 helix bundle, inducing its down/up movement (Supplementary Movie 2 and Supplementary Fig. 3g). This S2 helix bundle movement could even be propagated to the membrane-proximal stalk. Simultaneously, the RBD-2 tilts up/downward slightly, leaving/contacting the underneath S2 central helix of protomer 1 (Supplementary Movie 2), suggesting a compensate mode of motion between RBD-2 and RBD-3. Collectively, for Beta S trimer, the S1 subunit RBD-3 movement could be transferred to the central S2 helix bundle through contacting the HR1-CH hairpin, accordingly, S2 could also be involved in the collective motions with S1 movement, not seen in Kappa or WT counterpart[51]. Overall, this motion displayed by the Beta S trimer is distinct from the "breath" motion of the Kappa S, in which the three NTD-RBD pairs tilt outward/downward simultaneously, making the entire S trimer appear untwisted and expanded (Supplementary Movie 1 and Fig. 2g, h).

**Cryo-EM structure of the SARS-CoV-2 Kappa S-ACE2 complex.** Our BLI analysis showed a higher S/ACE2-binding affinity for the Kappa and Beta S than that of the WT and G614 S (Fig. 1a). To uncover the structural mechanism underlying the observed binding property changes, we carried out the cryo-EM analysis of the Kappa S trimer in complex with human ACE2 PD domain (Supplementary Fig. 4a–c, e). Four cryo-EM maps of the Kappa S trimer engaged with ACE2, including Kappa S-ACE2-C1 (only RBD-1 up), S-ACE2-C2a (RBD-1 and RBD-2 up), S-ACE2-C2b (RBD-1 and RBD-3 up), and S-ACE2-C3 (all three RBDs up), were determined at 4.0-, 3.9-, 3.9-, and 3.9-Å resolution, respectively (Fig. 4a–d, Supplementary Fig. 4a–c, and Supplementary Table 1). The models for the four structures were built accordingly (Supplementary Fig. 5a). We also focus-refined the stably associated RBD-1-ACE2 region to 3.8-Å resolution (Fig. 4e, f and Supplementary Fig. 4d). In Kappa S-ACE2-C1, engagement with ACE2 induces an upward tilt of RBD-1 relative to the S-surface from 72.5° to 77.1° (Fig. 2e). The Kappa S-ACE2-C1 complex generally overlaps with the WT S-ACE2 complex, yet appears a bit less compact, with RBD-2/3 slightly tilting upward and NTD-2 tilting outward a little (Supplementary Fig. 5b). Noteworthy, for the Kappa variant, ACE2 binding induces a more pronounced S-trimer population shift than that seen in the WT S-ACE2 system[8], suggesting that the Kappa S-ACE2 complex is prone to transforming to the more open C2a/C2b (51.7% population) and the fully open C3 (34.1%) states. RBD in the up position reduces the interaction between S1 and S2, and specifically releases the constraints imposed on the HR1-CH hairpin, which is known to completely refold during the membrane fusion process[52,53]. Thus, the more open spike (with more "up" RBDs) would be beneficial for the transformation of the S trimer toward the postfusion state and the simultaneous shedding of S1[8,10,54–56].

Further inspection of the Kappa RBD-1-ACE2 interaction interface revealed that the RBM T470-F490 loop, which plays vital roles in the engagement of SARS-CoV-2 spike with host-cell receptor ACE2 and with potent neutralizing MAbs[8,23,31], exhibits an observable outward shift of 1.7 Å for the Cα of V483 (Fig. 4f, g). Detailed inspection showed that the original negatively charged E484 could form contact with ACE2 K31. However, the substitution of the E484 with polar but uncharged Gln (Q) could reduce the original constrain, leading to the outward shift of the T470-F490 loop (Fig. 4g–i and Supplementary Table 3). For the L452R mutation, the hydrophobic to positively charged substitution also reverse the local surface property, with elongated

sidechain, thus affecting the binding of MAbs targeting this region (Fig. 4g–i).

**Cryo-EM structure of the SARS-CoV-2 Beta S-ACE2 complex.** We also examined the conformational space of the Beta S trimer in a complex with human ACE2. We determined four cryo-EM maps of the Beta S trimer engaged with ACE2, including Beta S-ACE2-C1 (only RBD-1 up), S-ACE2-C2a (RBD-1 and RBD-2 up), S-ACE2-C2b (RBD-1 and RBD-3 up), and S-ACE2-C3 (all three RBDs up), at 4.0-, 3.7-, 3.6-, and 3.6-Å resolution, respectively (Fig. 5a–d, Supplementary Fig. 6a–c, e, and Supplementary Table 2). The models for the four structures were built accordingly (Supplementary Fig. 7). We further focus-refined the stably associated RBD-1-ACE2 region to 3.9-Å resolution (Fig. 5e and Supplementary Fig. 6a, c, d). Overall, the pattern of these structures is similar to that observed in the Kappa S-ACE2 complex (Fig. 4a–d). In the S-ACE2-C2a, -C2b, and -C3 maps for both the Beta and the Kappa variants, the up RBD-2 or RBD-3 appear also to associate with ACE2 if we lower the map rendering threshold (Supplementary Figs. 4f and 6f), indicating a dynamic on/off of ACE2 association with these RBDs, in line with our BLI data showing relatively rapid disassociation kinetics between ACE2 and the S trimers ($k_{off} = 1.00 \times 10^{-2}$ s$^{-1}$/$9.34 \times 10^{-3}$ s$^{-1}$ for the Beta and Kappa S trimer, respectively; Fig. 1a). Similar to Kappa variant, ACE2 binding to Beta S also induces a pronounced S-trimer population shift, facilitating the transformation of the S-ACE2 complex to the more open C2a/C2b (64.0%) or fully open C3 (27.7%) states (Supplementary Fig. 6b), beneficial for the shedding of S1 and subsequent transformation toward the postfusion state.

In the Beta S-ACE2 structures, engagement of RBD-1 with ACE2 induces an upward tilt of RBD-1 relative to the S-surface from 70.8° to 76.3° (Fig. 2e). We then used the focus-refined RBD-1-ACE2 structure to further analyze their interaction interface (Fig. 5e–g). For the Beta variant, the original salt bridge formed between RBM K417 and ACE2 D30 was abolished due to the replacement of the positively charged K417 by a small uncharged Asn (N) (Fig. 5g, h and Supplementary Table 4). Still, the N501Y mutation could compensate for this loss by generating a new hydrogen bond between Y501 and K353 on ACE2 (Fig. 5g and Supplementary Table 4). The N501Y substitution could also form more aromatic interactions with neighboring residues including Y505 from RBD and the Y41 of ACE2 (Fig. 5g), in line with other reports[10]. This could contribute to the higher ACE2-binding affinity of the Beta S trimer relative to those of the WT and G614 S (Fig. 1a).

**ACE2 binding induced conformational dynamics of the Kappa and Beta S trimers.** We further examined the ACE2 binding induced conformational dynamics of the Kappa and Beta S trimers through 3DVA (Fig. 6 and Supplementary Movies 3–5). For the Kappa S trimer, upon ACE2 association, mode 1 displays a motion in which the up RBD-1-ACE2, together with the associated NTD-3, swings toward RBD-2 in an angular range of 9.5°, in a direction also observed in the WT S-ACE2 complex[8]. In the meanwhile, RBD-2 swings away from the center and RBD-3 swings in towards RBD-2 (Fig. 6a, b and Supplementary Movie 3). This motion presents conformational transitions between Kappa S-ACE2-C2a and -C2b states with RBD-1-ACE2 in the up position, and RBD-2 or -3 tilting up alternatively (Fig. 4b, c). Moreover, with the gradually enhanced association of ACE2 with RBD-1, mode 2 displays a motion in which RBD-1-ACE2 swings towards NTD-3, also observed in the WT S-ACE2 complex[8]. This RBD-1-ACE2 swing motion could reduce the constrains between RBD-1/2 and then between

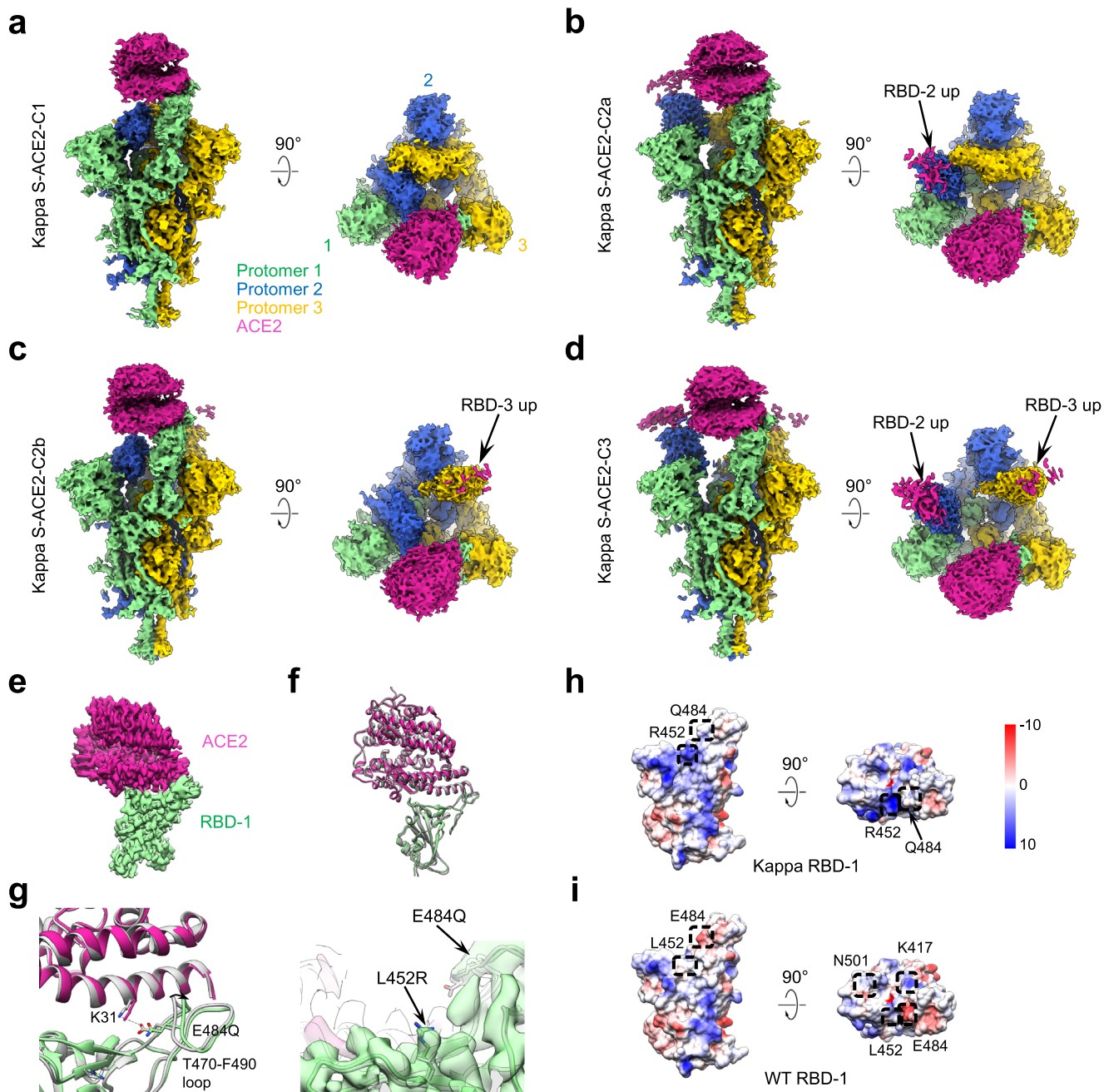

**Fig. 4 The conformers of the Kappa S-ACE2 complex and the interaction interface between the mutant RBD and ACE2. a–d** Four cryo-EM maps of the Kappa S-ACE2 complex in distinct conformations. ACE2 are shown in violet red. This color scheme is followed throughout. **e** Density map of the local refined Kappa RBD-1-ACE2. **f** Conformational comparison between Kappa RBD-1-ACE2 (in color) and WT RBD-ACE2 (PDB:6M0J, gray). **g** Zoom-in view of the Kappa RBD-ACE2 interface with substituted residues and relevant residues shown in stick (left). Right, model-map fitting shows the elongated sidechain of L452R is well resolved. **h, i** Surface properties of the Kappa RBD (**h**) and WT RBD (**i**). The mutated residues of RBD are marked in the black box.

RBD-2/3, consequently both RBD-2 and RBD-3 progressively tilt up, leading to the all-RBD-up conformation of Kappa S-ACE2-C3 (Fig. 6c, d and Supplementary Movie 4). Collectively, this motion exhibits conformational transitions from Kappa S-ACE2-C1 to -C3 state.

As for the Beta S-ACE2 complex, our 3DVA showed that the associated RBD-1-ACE2 exhibits a continuous swing motion leaving RBD-2 in an angular range of 9.4° (Fig. 6e, f), disturbing the original constrains between RBD-1 and RBD-2, leading to a slight upward tilt of RBD-2/3 in the transition from the Beta S-ACE2-C1 towards -C3 state. Overall, the S1 displays an expansion motion. Such

conformational dynamics of the fusion machinery with loosened constrain between the protomers could facilitate the shedding of S1 and transformation to the postfusion state.

In summary, our 3DVA data on the free S trimer of both Beta and Kappa variants suggested that the "up" RBD-1 swings approaching/leaving RBD-2, and the "down" RBD-2/RBD-3 could slightly tilt up, which are the intrinsic property of the S trimer of the two variants (Figs. 2g, h and 3f, g and Supplementary Movies 1 and 2). Once ACE2 associates with RBD-1, the motion of the engaged ACE2-RBD-1 appears in a larger scale (~9.4–9.5°) relative to that of the "up" RBD-1 in the

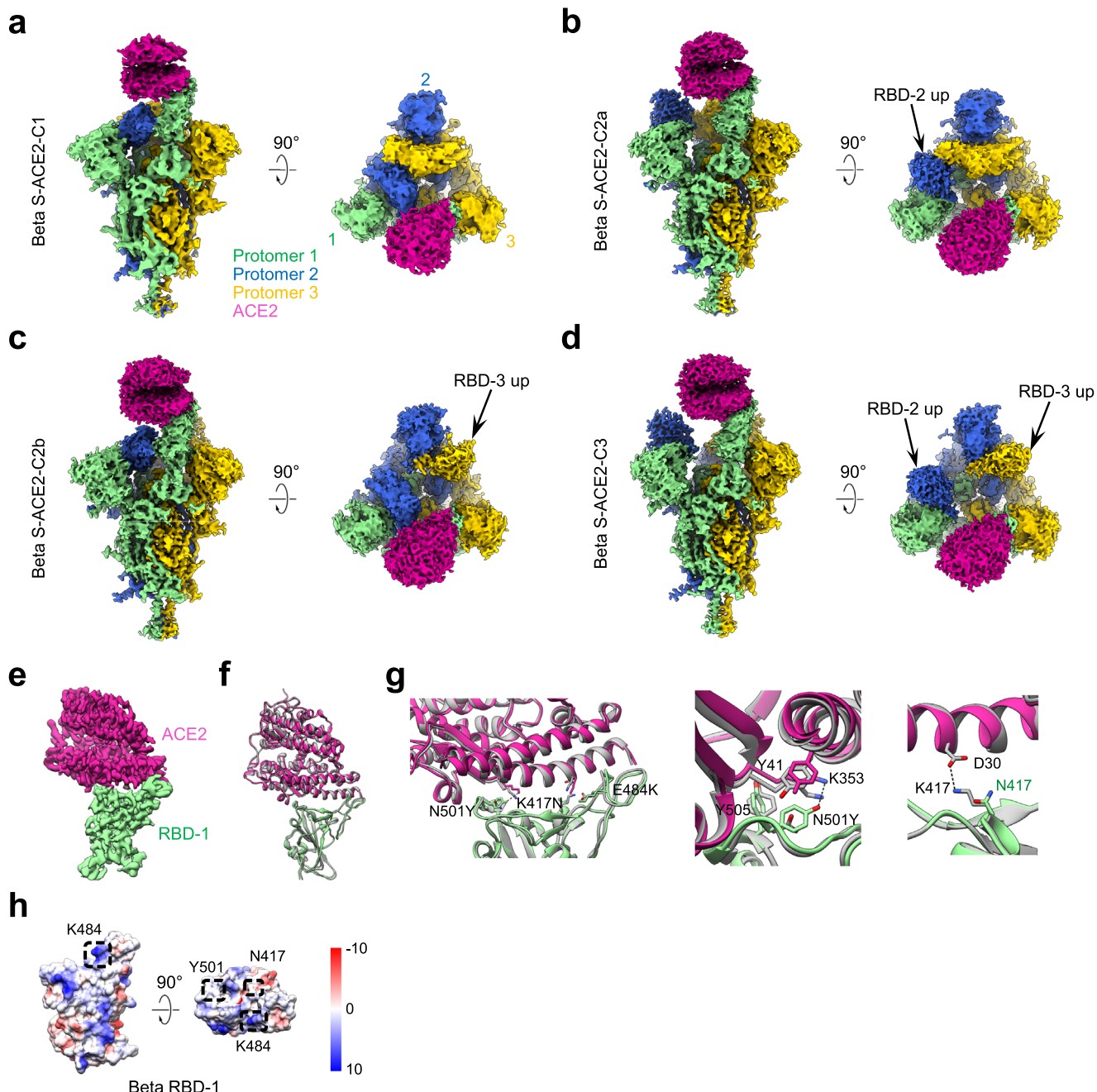

**Fig. 5 The conformers of the Beta S-ACE2 complex and the interaction interface between the mutant RBD and ACE2. a–d** Cryo-EM maps of the Beta S-ACE2 complex in different conformations. **e** Density map of the local refined Beta RBD-1-ACE2. **f** Conformational comparison between Beta RBD-1-ACE2 (in color) and WT RBD-ACE2 (PDB:6M0J, gray). **g** Zoom-in views of the Beta RBD-ACE2 interface with substituted residues and relevant residues shown in a stick. **h** Surface properties of the Beta variant RBD. The mutated residues of RBD are marked in the black box.

free S trimer (~6.2–8.0°), and the RBD-2/RBD-3 could swing to the fully "up" position (Fig. 6 and Supplementary Movies 3–5), illustrating ACE2 binding induced extra conformational dynamics of the S trimer.

## Discussion

The emergence of SARS-CoV-2 Beta and Kappa variants has created new challenges for the control of the ongoing COVID-19 pandemic. Understanding the nature and consequences of these variants is of significant importance. Here we determined two conformational states, including the S-open and the open-prone

S-transition, for each of the Kappa and Beta variant S trimers (Figs. 2 and 3). Compared to the WT S (94% particles in the tightly closed state and only 6% in the open state)[8] or the G614 S (35.6% open, 26.0% transition, and 38.5% closed)[43], the Kappa and Beta variants display a great population distribution shift towards the open state (~50% open-prone transition state and ~50% open state, without detecting tightly closed state) (Supplementary Figs. 2 and 3). Moreover, the S-open states for both variants appear more untwisted/open especially for the Beta S compared with those of the WT or G614 S (Figs. 2f and 3d, e). Interestingly, our 3DVA data suggested pronounced conformational dynamics for both Kappa and Beta S trimers especially in

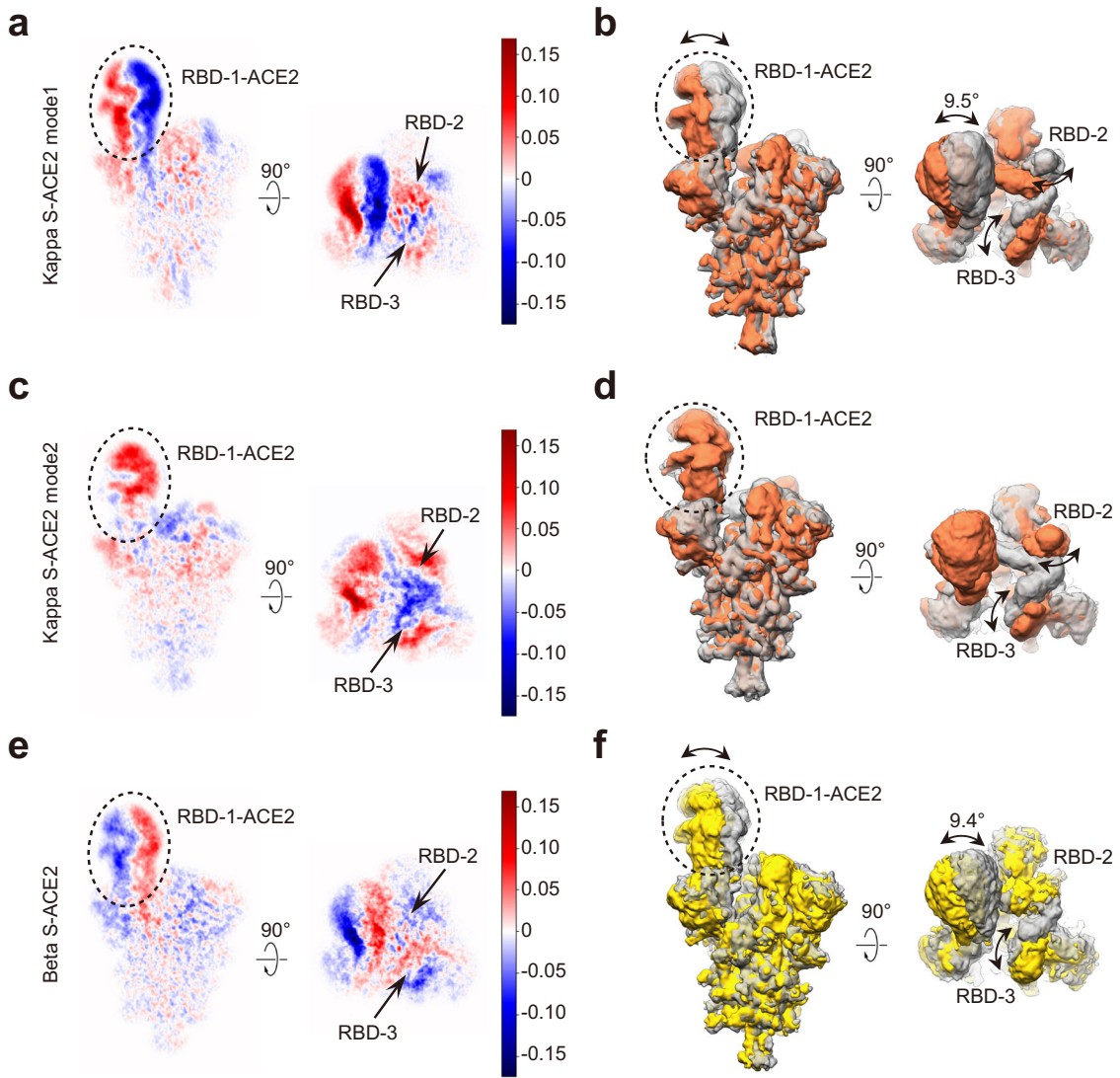

**Fig. 6 3D variability analysis of the Kappa S-ACE2 and Beta S-ACE2 complexes. a** Central slices of the 3DVA motion mode 1 of the Kappa S-ACE2 complex. Positive (red) and negative (blue) values correspond to density to be added and subtracted from the mean density. **b** The same motion displayed in maps showing two extreme conditions, with the angular range and direction shown. **c, d** The 3DVA motion mode 2 of the Kappa-S-ACE2 complex displayed in central slices (**c**) and in maps (**d**). **e, f** The representative 3DVA motion of the Beta S-ACE2 complex displayed in central slices (**e**) and in maps (**f**).

their S1 subunit, and a unique motion of the Beta S trimer with S2 potentially involved in the collective motion of S1 movement (Figs. 2g, h and 3f, g and Supplementary Movies 1 and 2). All these render the Kappa and Beta S more prone to receptor binding, which may increase their ACE2-binding affinity and ultimately lead to improved viral fitness and transmissibility. In fact, our BLI assays showed that the $K_D$ values of the Kappa and Beta S to monomeric ACE2 were 84 and 83 nM, respectively, lower than that of the parental G614 S ($K_D$ = 365 nM) (Fig. 1a). Our affinity data are in general consistent with those generated using a similar BLI protocol[30,43,46]. For example, Cai et al. have reported that the affinity of the Beta S to monomeric ACE2 ($K_D$ = 71.4 nM) increased as compared to that of the G614 S ($K_D$ = 124.0 nM)[30]. It should be mentioned that, for a given variant S protein, its absolute $K_D$ values generated from different studies[29,30,42–46], including two studies recently reported on BioRxiv[57,58], may vary significantly, likely due to multiple factors such as the method used (BLI or SPR), S protein construct (full-

length S or prefusion-stabilized S ectodomain), ACE2 form (monomeric or dimeric), the protein used to load the sensors (ACE2 or S), and even binding temperature. For example, it has been reported that the full-length G614 S had a $K_D$ value of 12.8 nM to dimeric ACE2 but its $K_D$ value to monomeric ACE2 was 124 nM[30]. Hence, it will be more meaningful to use the same assay system to compare different variant S proteins for affinity change.

Furthermore, we captured four distinct conformational states of the S trimer/ACE2 complex for each of the Kappa and Beta variants (Figs. 4 and 5). Combined with 3DVA results (Fig. 6 and Supplementary Movies 3–5), we were allowed to glimpse the main features of the dynamic process of conformational transitions induced by ACE2 binding, i.e., from the only up RBD-1 bound with an ACE2 to all three RBDs up with RBD-2/3 also partially engaged with ACE2 (Figs. 4–6 and Supplementary Figs. 4f and 6f). We have previously shown that for the WT S-ACE2 complex, the dominant population (73.8%) are in the

one RBD-up C1 state while the data did not depict multiple-RBD-up conformation[8]. In contrast, for Kappa and Beta S-ACE2 complexes, they all present pronounced population shift, i.e., with only 14.1% or 8.3% population in the one RBD-up C1 state while the remaining dominant population existing in the more RBM exposed C2a/C2b and C3 states with two or three RBDs up and possibly engaged with more ACE2s (Supplementary Figs. 4b, f and 6b, f). Transformation to the fusion-prone C2a/C2b and C3 states can lead to loss of concealment for the S2 central helixes, which was offered by the RBDs resting above them, and decrease in S1/S2 interaction, facilitating the shedding of S1 and transformation to the postfusion state. We also found that, for Kappa S, the E484Q substitution modifies the surface property, leading to a small outward shift of the T470-F490 loop which may affect the interaction with ACE2 (Fig. 4g). For the Beta variant, although K417N substitution abolishes a salt bridge between K417 and ACE2 D30 (Fig. 5g), the N501Y could compensate this loss by forming a new hydrogen bond with K353 on ACE2, and more aromatic interactions with the neighboring Y505 from RBD and Y41 of ACE2 (Fig. 5g, h), resulting in slightly higher ACE2-binding affinity relative to those of the WT and G614 S (Fig. 1a). Interestingly, we find that, upon ACE2 binding, S-ACE2 complexes for both variants all exhibit an enhanced swing motion in RBD-1-ACE2, potentially perturbing the original constrains among the RBDs, leading to more RBDs opened up and larger global motions especially in the S1 region. Collectively, for the Beta and Kappa S trimers, these surface property changes, extra conformational dynamics, and population shift could together reshape their conformational landscape, rendering the fusion machinery capable of receiving more receptors and potentially lowering the energy barrier of their transformation to the postfusion state, ultimately leading to enhanced infectivity/transmissibility of the Beta and Kappa variants.

It has been shown that Beta is resistant to some potent neutralizing MAbs to original strains[59,60]. In this study, we found that 2H2, a potent neutralizing MAb that binds the RBD of WT SARS-CoV-2[23], lost neutralization potency towards Beta and Kappa variants (Fig. 1c). Consistently, drastically diminished or abrogated 2H2 binding was observed for Beta and Kappa-S trimers which carry three (K417N, E484K, and N501Y) and two (L452R and E484Q) mutations in their RBD regions, respectively. These mutation sites are all located in the binding footprint of 2H2 on the WT RBD (Fig. 1e)[23]. Further analysis using RBD mutants showed that 2H2 failed to bind the RBDs containing E484K or triple mutations (K417N, E484K, and N501Y) (Fig. 1d), indicating that E484 is critical for RBD binding with 2H2. This result is in well agreement with our previous structural finding that the WT RBD V483-F490 loop forms intense contacts with the CDRL1, CDRL3, CDRH2, and CDRH3 loops of 2H2 and in particular E484 forms contact simultaneously with W52, R53, N98, and H102 of 2H2 heavy chain[23]. The Beta S trimer structures from this study show that the E484K mutation changes the 2H2 binding footprint (epitope) property from acidic to basic (Figs. 5h and 4i). Similarly, for the Kappa variant, the E484Q substitution reverts the surface property (Fig. 4h, i), and in addition, the L452R mutation (consecutive to the Y453-L455 region) results in a change of the nonpolar L452 to positively charged R with an enlarged sidechain (Fig. 4h, i). Hence, besides the loss of E484-based direct contacts, the changes in surface property and size of the sidechain, as well as conformational variations seen in the Beta and Kappa S trimers may also disturb the interaction between their RBD and 2H2, contributing to the loss of 2H2 binding and resistance to 2H2 neutralization.

In summary, by determining an ensemble of cryo-EM structures of the Kappa and Beta S trimers and their complexes with

ACE2 receptor, we reveal enhanced conformational dynamics and population shift towards the all RBD-up fusion-prone open states for Kappa and Beta, implicating that mutations in these variants may modify the kinetics of receptor binding and viral fusion to improve virus fitness. In addition, we found that the variants efficiently interact with ACE2 receptor despite their sensitive ACE2-binding surface is modified to escape recognition by some potent neutralizing MAbs directed to RBM. These findings shed new light on the pathogenicity and immune evasion mechanism of the Beta and Kappa variants, providing important information for the control of the SARS-CoV-2 pandemic.

## Methods

**Expression and purification of SARS-CoV-2 variants Kappa and Beta S and human ACE2.** Prefusion-stabilized SARS-CoV-2 S trimer and human ACE2 were generated in a previous study[8]. Briefly, the mammalian codon-optimized gene coding SARS-CoV-2 (Wuhan-Hu-1 strain, GenBank ID: MN908947.3) S glycoprotein ectodomain was cloned into vector pcDNA 3.1 +, with proline substitutions at K986 and V987, a "GSAS" substitution at the furin cleavage site (R682 to R685). A C-terminal T4 fibritin trimerization motif, a TEV protease cleavage site, a FLAG tag, and a His tag were cloned downstream of the SARS-CoV-2 S glycoprotein ectodomain. A gene encoding human ACE2 PD domain (Q18-D615) with an N-terminal interleukin-10 (IL-10) signal peptide and a C-terminal His tag was cloned into vector pcDNA 3.4. To prepare prefusion-stabilized S proteins of SARS-CoV-2 G614, Beta (hCoV-19/South Africa/KRISP-BH02956751/2020, GISAID ID: EPI_ISL_736940) and Kappa (hCoV-19/India/WB-1931500939910/2021, GISAID ID: EPI_ISL_1589917) variants, D614G amino acid substitutions of G614, mutations of Beta (L18F, D80A, D215G, DEL242-244, R246I, K417N, E484K, N501Y, D614G, A701V) and mutations of Kappa (E154K, L452R, E484Q, D614G, P681R) were induced by site-directed mutagenesis, using prefusion-stabilized SARS-CoV-2 S-trimer expression plasmid. Primers used in this study are provided in Supplementary Table 5. Note that the sequence of the Kappa variant adopted here is derived from an earlier Kappa strain and hence does not contain the G142D and Q1071H mutations compared to the Kappa variant sequence defined later by WHO (https://www.cdc.gov/coronavirus/2019-ncov/variants/variant-info.html). The proteins were purified according to the published protocol[8]. Briefly, the vectors were transiently transfected into HEK293F cells (Thermo Fisher) using polyethylenimine. Three days after transfection, the supernatants were harvested, the clarified supernatants were added with 20 mM Tris-HCl pH 7.5, 200 mM NaCl, 20 mM imidazole, 4 mM MgCl$_2$, and incubated with Ni-NTA resin at 4 °C for 1 h. The Ni-NTA resin was recovered and washed with 20 mM Tris-HCl pH 7.5, 200 mM NaCl, 20 mM imidazole. The protein was eluted by 20 mM Tris-HCl pH 7.5, 200 mM NaCl, 250 mM imidazole.

**Biolayer interferometry (BLI) assay.** To determine binding affinity of ACE2, SARS-CoV-2 S trimer and the variants proteins were first subjected to gel filtration chromatography using a Superose 6 increase 10/300 GL column (GE Healthcare) pre-equilibrated with PBS and then biotinylated using the EZ-Link™ Sulfo-NHS-LC-LC-Biotin kit (Thermo Fisher) and purified using Zeba™ spin desalting column (Thermo Fisher). Biotinylated SARS-CoV-2 S trimer and variants proteins were loaded onto streptavidin (SA) biosensors (Pall FortéBio). The biosensors were dipped into wells containing varying concentrations of ACE2 protein. For WT S, Kappa S and Beta S, ACE2 concentration range used was 200–0.823 nM, while for G614S variant, ACE2 concentration range was 1800 to 7.41 nM, since at ACE2 concentration of 7.41 nM, the signal value was already close to 0. The interactions were monitored over a 500-s association period. Finally, the sensors were switched to dissociation buffer (10 mM PBS, 0.02% Tween 20 and 0.1% bovine serum albumin) for a 500-s dissociation phase. The data were corrected by subtracting the reference sample and then fitted to a 1:1 binding model for the determination of affinity constants using the software Octet Data Analysis 11.0.

**Binding of S trimers with MAbs analyzed by ELISA.** To evaluate the binding properties of the S proteins of the variants, ELISA plates were coated with serially diluted wild-type, Beta or Kappa S trimers (50 μL/well) at 37 °C for 2 h, followed by blocking with 5% milk in PBS-Tween 20 (PBST). After washes, the plates were incubated with 50 ng/well of the MAbs 3C1 or 2H2[23] at 37 °C for 2 h, followed by horseradish peroxidase (HRP)-conjugated anti-mouse IgG (Sigma, diluted 1:10,000). After washing and color development, absorbance was monitored at 450 nm.

**Murine leukemia virus (MLV)-based pseudovirus-neutralization assay.** MLV-based SARS-CoV-2 S pseudoviruses were generated according to our previously reported method[23] with few modifications. Briefly, the plasmids coding the full-length S proteins of SARS-CoV-2 Kappa or Beta strains were constructed and used for the production of the corresponding pseudoviruses.

Pseudovirus-neutralization assay was performed with human ACE2-expressing HEK 293T cells (293T-hACE2) following our previously described method[23]. At 48 h post infection, luciferase activity was measured and the percentage of neutralization was calculated. For each MAb, half inhibitory concentration ($IC_{50}$) was calculated using nonlinear regression in GraphPad Prism (version 8).

**Effects of RBD mutations on MAb binding determined by ELISA**. To assess the influence of naturally occurring RBD mutations on the binding of anti-RBD MAbs[23], a series of single-point and triple-SARS-CoV-2 RBD mutants were constructed, including RBD (K417N), RBD (E484K), RBD (N501Y), and Beta-RBD (carrying the K417N-E484K-N501Y mutations). Specifically, recombinant plasmids encoding these RBD mutants were generated based on the parental plasmid pcDNA 3.4-SARS-2-RBD[23] using the Mut ExpressTM II Fast Mutagenesis Kit V2 (Vazyme, China) following the manufacturer's instructions. The plasmids were transfected into HEK293F cells using PEI. At day 5, culture supernatants were collected and his-tagged mutant RBD proteins were purified by affinity chromatography with Ni-NTA resin (Millipore).

Binding of anti-RBD MAbs[23] to the purified RBD mutants was measured by ELISA. Briefly, 96-well ELISA plates were coated with individual RBD mutant (100 ng/well) in PBS and then blocked with 5% milk in PBS-Tween 20 (PBST). The plates were incubated with anti-RBD MAbs (50 ng/well), anti-zika virus MAb 5F8[61] (50 ng/well; control), or mouse anti-RBD polyclonal antibody (diluted at 1/1000) at 37 °C for 2 h. After washing, horseradish peroxidase (HRP)-conjugated anti-mouse IgG (Sigma) was added to the plates and incubated. After washes and color development, absorbance at 450 nm was measured.

**Cryo-EM sample preparation**. To prepare the cryo-EM sample of the Kappa and Beta SARS-CoV-2 S trimer, a 2.2 μL aliquot of the S sample (~3 mg/mL) was applied on a plasma-cleaned holey carbon grid (R1.2/1.3, Cu, 200 mesh; Quantifoil). The grid was blotted with Vitrobot Mark IV (Thermo Fisher Scientific) using a blot force of -1 and 1 s blot time at 100% humidity and 8 °C and then plunged into liquid ethane cooled by liquid nitrogen. To prepare the cryo-EM sample of Kappa/Beta S-ACE2 complex, purified S was incubated in a 1:4 molar ratio with ACE2 on ice for 20 min and then adopted the same vitrification procedure as for the S trimers.

**Cryo-EM data collection**. Cryo-EM movies of the samples were collected on a Titan Krios electron microscope (Thermo Fisher Scientific) operated at an accelerating voltage of 300 kV with a magnification of 64,000×. The movies were recorded on a K3 direct electron detector (Gatan) operated in the counting mode (yielding a pixel size of 1.093 Å) under a low-dose condition in an automatic manner using EPU software (Thermo Fisher Scientific). Each frame was exposed for 0.1 s, and the total accumulation time was 3 s, leading to a total accumulated dose of 50 e⁻/Å² on the specimen. To solve the problem of preferred orientation associated with Beta S trimer, we additionally collected tilt datasets with the stage tilt at 30°, while the other conditions remained the same.

**Cryo-EM 3D reconstruction**. For each dataset, the motion correction of the image stack was performed using the embedded module of Motioncor2 in Relion 3.1[62,63] and CTF parameters were determined using CTFFIND4[64] before further data processing. Unless otherwise described, the data processing was performed in Relion 3.1. For the Kappa S dataset, we obtained 1,405,927 particles by automatic particle picking and 464,337 particles remained after reference-free 2D classification. The cleaned-up particles were used for further reconstruction with the WT S-open map (EMD-21457) as the initial model[48]. After two rounds of 3D classifications, we obtained a Kappa S-open map from 70,747 particles and an S-transition map from 73,325 particles. After Bayesian polishing and CTF refinement, the Kappa S-open and S-transition maps were refined to 3.2- and 3.4-Å resolution, respectively. Finally, the Kappa-S-open and S-transition maps were post-processed by utilizing deepEMhancer[65]. We performed further 3D Variability analysis in cryoSPARC v3.2.0[50] to capture continuous conformational dynamics of Kappa S. The overall resolution was determined based on the gold-standard criterion using a FSC of 0.143. For the Beta S dataset, a similar data processing procedure was adopted as for the Kappa-S one.

For the Kappa S-ACE2 dataset, we obtained 2,837,416 particles by automatic particle picking and 336,804 particles remained after reference-free 2D classification and two rounds of 3D classifications. These particles were refined to 3.8 Å after Bayesian polishing and CTF refinement. We then further cleaned up the particles by applying focused 3D classification in RBD-1-ACE2 region and obtained 259,779 particles, which were further refined to 3.7-Å resolution for the complete complex. To improve the local resolution of the RBD-1-ACE2 region, we transferred these particles into cryoSPARC, and applied local refinement with a soft mask covering only the RBD-1-ACE2 region and obtained a 3.8-Å-resolution map of RBD-1-ACE2. To sort out conformational difference in the RBD-2/RBD-3 region, we did additional focused 3D classification on RBD-2/RBD-3 region in cryoSPARC, and obtained four different Kappa S-ACE2 conformational states, including C1 (36,511 particles), C2a (66,768 particles), C2b (67,543 particles), and

C3 (88,957 particles), which were non-uniform refined to 4.0-, 3.9-, 3.9-, and 3.9-Å resolutions, respectively. We also carried out 3DVA in cryoSPARC on the 259,779-particle dataset to capture the conformational dynamics of the Kappa S-ACE2 complex. For the Beta S-ACE2 dataset, we applied similar data processing procedures described above as for the Kappa S-ACE2 complex.

**Pseudoatomic model building**. To build the pseudoatomic models for our Kappa and Beta S-open structures, we used the available atomic model of SARS-CoV-2 S-open (PDB: 7DK3) as initial model[8]. We first fit the model in the corresponding cryo-EM map in Chimera by rigid-body fitting, and manually substituted the mutations of the Kappa and Beta variants in COOT[66]. Then we used Rosetta to refine the models against the density map[67], and eventually used phenix.real_space_refine for S-trimer model refinement against the corresponding map[68]. For Kappa and Beta S-transition structures, we utilized the SARS-CoV-2 S models (PDB: 7DK3, 7KRQ, 7KRS) as initial templates and followed similar procedure as for the S-open state[8,43]. For the Kappa S-ACE2 and Beta S-ACE2 structures, we used the SARS-CoV-2 S-ACE2 models (PDB: 7DF4)[8] as initial templates, the other steps were performed in the same way as mentioned above. For Kappa and Beta RBD-1-ACE2 structures, we used the SARS-CoV-2 RBD-ACE2 crystal structure (PDB: 6M0J) as an initial model[4], and applied phenix.real_space_refine for the model refinement against the corresponding maps. The final pseudoatomic models were validated using Phenix.molprobity command in Phenix. Interaction surface analysis was conducted by utilizing PISA server[69].

UCSF Chimera and ChimeraX were applied for figure generation, rotation measurement (by using "measure rotation" command), and coulombic potential surface analysis[70,71]. Some studies use centroids of domains to define vectors and measure the relative motion of a certain domain between two conformers, in a way somewhat different from our rotation measurement, while their choice of pivot points is close to the position of our rotation axis for NTD and RBD[72,73].

**Reporting summary**. Further information on research design is available in the Nature Research Reporting Summary linked to this article.

## Data availability

All data presented in this study are available within the figures and in the Supplementary Information. For the SARS-CoV-2 Kappa variant, related cryo-EM maps have been deposited at the Electron Microscopy Data Bank with accession codes EMD-32177, EMD-32180, EMD-32172, EMD-32173, EMD-32174, EMD-32175, and EMD-32169, and associated atomic models have been deposited in the Protein Data Bank with accession codes 7VXE, 7VXI, 7VX9, 7VXA, 7VXB, 7VXC, and 7VX5 for S-open, S transition, C1, C2a, C2b, C3, and RBD-1-ACE2, respectively. For the SARS-CoV-2 Beta variant, related cryo-EM maps have been deposited at the Electron Microscopy Data Bank with accession codes EMD-32167, EMD-32170, EMD-32176, EMD-32182, EMD-32178, EMD-32184, and EMD-32168, and associated atomic models have been deposited in the Protein Data Bank with accession codes 7VX1, 7VX7, 7VXD, 7VXK, 7VXF, 7VXM, and 7VX4 for S-open, S transition, C1, C2a, C2b, C3, and RBD-1-ACE2, respectively. The structures were used for initial templates or structural analysis in this work including PDB IDs: 7DK3, 7KRQ, 7KRS, 7KRR, 7DF4, 7DK4, 7DCC, and 6M0J. Source data are provided with this paper.

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

## Acknowledgements

We are grateful to the staffs of the NCPSS Electron Microscopy facility, Database and Computing facility, and Protein Expression and Purification facility for instrument support and technical assistance. This work was supported by grants from the Strategic Priority Research Program of CAS (XDB37040103 to Y.C. and XDB29040300 to Z.H.), National Key R&D Program of China (2017YFA0503503 to Y.C. and 2020YFC0845900 to Z.H.), the NSFC (31670754 and 31872714 to Y.C.), the NSFC-ISF 31861143028, Shanghai Academic Research Leader (20XD1404200 to Y.C.), and the CAS Facility-based Open Research Program and the CAS-Shanghai Science Research Center (CAS-SSRC-YH-2015-01, DSS-WXJZ-2018-0002 to Y.C.). This project is part of the European Union's Horizon 2020 research and innovation program under grant agreement No 101003589 to Z.H. This work was also supported by China National Postdoctoral Program for Innovative Talents (BX2021310 to C.X.), the Youth Innovation Promotion Association of the Chinese Academy of Sciences (CAS) and Shanghai Rising-Star Program (21QA1410000 to C.Z.).

## Author contributions

Y.C. and Z.H. designed the experiments; Y.-X.W., Z.L., and S.X. expressed and purified the proteins; Q.H. and Y.-F.W. performed the cryo-EM acquisition; Y.-F.W. performed the cryo-EM reconstructions with the involvement of Q.Z. in particle picking; C.X. performed the model building; Y.-X.W. and C.Z. performed the biochemical analyses; C.L. made the movies; Y.C., Z.H., Y.-F.W., C.X., Y.-X.W., and C.Z. analyzed the data and wrote the manuscript with inputs from all the others.

## Competing interests

The authors declare no competing interests.
