## [Peer Review File · Nature Communications]

Conformational dynamics of the Beta and Kappa SARS-CoV-2 spike proteins and their complexes with ACE2 receptor revealed by cryo-EMREVIEWER COMMENTS

Reviewer #1 (Remarks to the Author):

SARS-CoV-2 variants of higher transmissibility and infectivity are emerging continuously, further aggravating the pandemic of COVID-19. Among these variants, the lineages B.1.1.7, B.1.351, P.1, and B.1.617.2 are variants of concerns and named as Alpha, Beta, Gamma, and Delta variants by WHO respectively. The lineage B.1.617.1 are variant of interest and named as Kappa by WHO. All of these variants mentioned above carry dozens of mutations, especially in the S protein that play important role during virus infection. The structure of the Kappa or the Delta variant S protein is not reported, though the structures of the Alpha, Beta and Gamma variant S protein have been published. In this work, the authors found that both the Beta and Kappa variants can escape from the neutralization by the MAb 2H2, but not 3C1. The authors solved high resolution cryo-EM structures of the S protein of the Beta and Kappa variants in alone or in complex with the ACE2 receptor, revealing the structural basis for the higher infectivity and the immune escape of these two virus strains. Here are several points to be addressed for this work.

1. WHO have named the major variants of SARS-CoV-2. Please introduce and use these names.
2. The authors performed 3DVA on the S variant proteins alone or in complex with ACE2 to reveal the conformational dynamics. The 3DVA movies show that the conformational changes are very similar for the S protein alone and in complex with ACE2. How do the authors discriminate the conformational dynamics induced by ACE2 binding and by the intrinsic property of the S protein?
3. The authors revealed a transition state for both variant S proteins. What is the difference between the "transition" state and the closed state described in the previous works?
4. Please label key domains on the 3DVA movies.
5. Please specify the virus name and the GISAID ID for the variant S protein used in this work.

Reviewer #2 (Remarks to the Author):

Wang et al. reported a comprehensive study of the structural and biophysical characterizations of the spike protein of the SARS-CoV-2 B.1.351 (Beta) and B.1.617.1 (Kappa) variant. Using the same workflow as described previously (doi: 10.1126/sciadv.abe5575 & 10.1038/s41467-020-20465-w), the authors identified different number of RBDs in the up conformation for the B.1.351 and B.1.617.1 spike variants in the absence and presence of ACE2. Through the 3D variability analysis, the authors were able to quantify the relative populations of the RBD-up conformations and the corresponding conformational variations expressed in terms of the rotations of the individual domains. Although the two variants are not as problematic as the Delta (B.1.617.2) variant in terms of the number infection cases and death rate on the global scale, they nevertheless pose serious challenges to the COVID-19 health care system because of their reduced sensitivity to vaccine-elicited sera and monoclonal neutralizing antibodies. The structural and functional information derived from this study should therefore provide useful molecular insights to the understanding of host recognition and immunity evasion of the emerging SARS-CoV-2 variants. There are however, a number of important issues that should be addressed for a major revision. In particular, discrepancies between the current study and literature reports on the relatively ACE2 binding affinities between different variants should be carefully addressed.

Major comments:

1. The construct of B.1.617.1 as specified in Fig. S1A differs from the commonly used definition of the B.1.617.1 (<https://www.cdc.gov/coronavirus/2019-ncov/variants/variant-info.html>) in that G142D and Q1071H are missing in this study. The authors ought to clarify why these mutations are not included in the construct, and how these differences may impact on the structure and function of the spike protein.
2. The authors used BLI analyses and WT-S as a reference to compare the ACE2 binding affinities of individual spike variants. Fig. 1A showed a three-fold reduction in ACE2 binding affinity for G614-S, a

50% reduction for B.1.617.1-S and a two-fold increase for B.1.351-S. First of all, the authors only presented the raw sensorgrams without showing the fitting results, making it difficult to evaluate the quality of the experimental data. Some flaw in the data quality is already evident as the dissociation curves often overlap (WT-S and G614G-S) and the spacing between different sensorgrams are not appropriately distributed according to the specified protein concentrations. Furthermore, the authors did not specify the construct design of ACE2 in the Method section in which the expression and purification of recombinant spike variants and ACE2 are described. These technical problem raise the issue of the validity of the BLI analysis.

3. Tokunaga and co-workers recently reported an increase of ACE2 binding for the D614G variant, which contrasts the opposite results as described herein. Using the same BLI analysis, their K_d values are two orders of magnitude lower than the values reported in this study. The difference in the absolute scales and the relative affinity between these two studies should be addressed.

4. Likewise, the BLI analyses reported by the two recent Science papers (10.1126/science.abf2303 & 10.1126/science.abi6226) on the B.1.351 variants and the two preprints (<https://doi.org/10.1101/2021.08.11.455956> and <https://www.biorxiv.org/content/10.1101/2021.08.17.456689v1>) on the Kappa variant should be commented. In particular, the Kappa variant was found to bind to ACE2 much more strongly than WT in both preprints, but the current study showed otherwise. The potential source of the contraction should be appropriately addressed.

5. With regard to the geometrical analysis of the RBD conformational changes as illustrated in Fig. 2D/F/H, Fig. 3D/E/G, and Fig. 6, the points of rotation (hinge) should be explicitly defined as the portions of the hinge can make a big difference in the results of the rotation angles. In particular, it is unclear whether the RBDs in Fig. 2H and Fig. 3G are rotating around the long axis defined by the central helix of the S2 subunit of out of plane whose normal is parallel to the long axis of the central helix, i.e., a upward motion as defined in Fig. 2D. The repeating 3DVA analysis results of Fig. 2G-H and Fig. 3F-G are somewhat superfluous, since they report the same features.

6. The functional implications of the RBD motions should be better elaborated. For example, in line 208, why would more open spike be beneficial for the shedding of S1? Appropriate reference should be included.

7. Discussion, first paragraph, "our 3DVA data suggested pronounced conformational dynamics especially for B.1.351 S trimer with S2 potentially involved in the allosteric cooperation in S1 movement." How can the 3DVA analysis inform us on allostery? More specifically, it is not very clear from Fig. 2G-H and 3F-G what allostery can be inferred from the motions. The authors should define the allostery in the context of spike dynamics with better schematic representations.

Minor comments:

1. The sentence in line 192 (Four cryo-EM map,...) is very long and hard to read. It should be rephrased.

2. The dissociation constant K_d should be subscript (Fig. 1A) and the on- and off-rates, k_{on} and k_{off} , should be lower case k with on and off (dis is an unusual usage for the off rate) as subscripts.

3. IC₅₀ in Fig. 1C (50 should be subscript, too)

4. Fig. S6. It may be the threshold issue that caused the loss of the ACE2 EM density in Fig. S6E C2 and C2b when three and two ACE2 are expected but only one is shown. This is contrasting the result in Fig. S6F in which three ACE2 are shown although two show weaker density.

REVIEWER COMMENTS

Reviewer #1

SARS-CoV-2 variants of higher transmissibility and infectivity are emerging continuously, further aggravating the pandemic of COVID-19. Among these variants, the lineages B.1.1.7, B.1.351, P.1, and B.1.617.2 are variants of concerns and named as Alpha, Beta, Gamma, and Delta variants by WHO respectively. The lineage B.1.617.1 are variant of interest and named as Kappa by WHO. All these variants mentioned above carry dozens of mutations, especially in the S protein that play important role during virus infection. The structure of the Kappa or the Delta variant S protein is not reported, though the structures of the Alpha, Beta and Gamma variant S protein have been published. In this work, the authors found that both the Beta and Kappa variants can escape from the neutralization by the MAb 2H2, but not 3C1. The authors solved high resolution cryo-EM structures of the S protein of the Beta and Kappa variants in alone or in complex with the ACE2 receptor, revealing the structural basis for the higher infectivity and the immune escape of these two virus strains. Here are several points to be addressed for this work.

--We thank the reviewer for the insightful comments.

Q1-1. WHO have named the major variants of SARS-CoV-2. Please introduce and use these names.

A1-1: The suggestion is well taken. We are aware that the World Health Organization (WHO) have updated the nomenclature of SARS-CoV-2 variants, e.g., "Alpha" for B.1.1.7, "Beta" for B.1.351, "Gamma" for P.1, "Kappa" for B.1.617.1, and "Delta" for B.1.617.2. As suggested, we have now added a brief introduction on the new naming system for major variants and adopted the new names in our revised manuscript (please see Page 3, Line 56-59).

Q1-2. The authors performed 3DVA on the S variant proteins alone or in complex with ACE2 to reveal the conformational dynamics. The 3DVA movies show that the conformational changes are very similar for the S protein alone and in complex with ACE2. How do the authors discriminate the conformational dynamics induced by ACE2 binding and by the intrinsic property of the S protein?

A1-2: Thanks for the comment from our reviewer. For the free Kappa S trimer, our 3DVA mainly displays a “breath” motion (Movie S1), with the three NTD-RBD pairs (including NTD-1/RBD-2, NTD-2/RBD-3 and NTD-3/RBD-1) tilting outward/downward simultaneously, consequently the entire S trimer appears untwisted and expanded (Movie S1, Fig. 2G-H). While for the free Beta S trimer, 3DVA data suggest that the RBD-3 movement, through contacting the underneath HR1-CH hairpin of the neighboring protomer, could be allosterically transformed to the central S2 helix bundle (Movie S2, Fig. 3F-G). Commonly, for the free S trimer of both Beta and Kappa variants, their “up” RBD-1 swing approaching/leaving RBD-2, and their “down” RBD-2/RBD-3 display observable movement slightly tilting up towards the transition state (Fig. 2H, 3G and Movie S1-S2). These are the intrinsic property of the S trimers.

In contrast, once ACE2 associates with RBD-1, the motion of the associated ACE2-RBD-1 appears in a larger scale ($\sim 9.4^\circ$ - 9.5°) relative to that of the “up” RBD-1 in the free S trimer ($\sim 6.2^\circ$ - 8.0°), and the RBD-2/RBD-3 could swing to the fully “up” position (Fig. 6 and Movie S3-S5), potentially allowing S trimer to engage with more ACE2. These are the ACE2-binding induced extra conformational dynamics. Corroborating to our 3DVA results, we only observe one “up” RBD for the free S trimer, while capture two or three “up” RBDs (S-ACE2-C2a/-C2b/-C3) in the S-ACE2 dataset for both Beta and Kappa variants. We have added an elaboration of this point in our revised manuscript (please see Line 275-282 on Page 10).

Q1-3. The authors revealed a transition state for both variant S proteins. What is the difference between the “transition” state and the closed state described in the previous works?

A1-3: In the closed state of SARS-CoV-2 S protein described in previous studies^{1,2}, the S trimer is highly symmetric with all the three RBDs buried in the “down” position (Fig. R1A, D). Here in the “transition” state of Beta and Kappa variants, with the RBD-1 slightly tilting up (Fig. R1B-C), the symmetry between the three protomers is broken, and a gap appears between the “up” RBD-1 and the “down” RBD-3, which could be observed in the top view (Fig. R1E, Fig. 2C and 3C). These features discriminate the transition state from the closed state.

Fig. R1 Structural comparison of the S trimer between transition and closed state. (A) Side view of the closed state of the G614 S (PDB:7KRQ, in hot pink)¹. (B) Side view of the transition state for Kappa and Beta variants (in color). (C) Structural comparison of the protomer 1 between the transition state (in light green) and the closed state (in hot pink), and the zoomed-in view showing the slightly tilting up of the RBD-1 and associated underneath SD1 motif. (D) Top view of the closed state of the G614 S trimer (PDB:7KRQ) shows highly symmetric feature between the three protomers. (E) Top view of the transition state of the variants shows a gap between the “up” RBD-1 and the “down” RBD-3.

Q1-4. Please label key domains on the 3DVA movies.

A1-4: The suggestion is well taken. We have regenerated the 3DVA movies and labeled the key structure elements including RBDs and NTDs that show obvious movements.

Q1-5. Please specify the virus name and the GISAID ID for the variant S protein used in this work.

A1-5: Thanks for the suggestion. The variants selected for S protein expression include Kappa (B.1.617.1) strain hCoV-19/India/WB-1931500939910/2021 (GISAID ID: EPI_ISL_1589917), and Beta (B.1.351) strain hCoV-19/South Africa/KRISP-BH02956751/2020 (GISAID ID: EPI_ISL_736940). We have now specified the virus name and the GISAID ID in our revised manuscript in the Method section (please see Page 13, Line 375-376).

Reviewer #2

Wang et al. reported a comprehensive study of the structural and biophysical characterizations of the spike protein of the SARS-CoV-2 B.1.351 (Beta) and B.1.617.1 (Kappa) variant. Using the same workflow as described previously (doi: 10.1126/sciadv.abe5575 & 10.1038/s41467-020-20465-w), the authors identified different number of RBDs in the up conformation for the B.1.351 and B.1.617.1 spike variants in the absence and presence of ACE2. Through the 3D variability analysis, the authors were able to quantify the relative populations of the RBD-up conformations and the corresponding conformational variations expressed in terms of the rotations of the individual domains. Although the two variants are not as problematic as the Delta (B.1.617.2) variant in terms of the number infection cases and death rate on the global scale, they nevertheless pose serious challenges to the COVID-19 health care system because of their reduced sensitivity to vaccine-elicited sera and monoclonal neutralizing antibodies. The structural and functional information derived from this study should therefore provide useful molecular insights to the understanding of host recognition and immunity evasion of the emerging SARS-CoV-2 variants. There are, however, a number of important issues that should be addressed for a major revision. In particular, discrepancies between the current study and literature reports on the relatively ACE2 binding affinities between different variants should be carefully addressed.

--- Thanks for the encouraging comments and constructive suggestions from our reviewer.

Major comments:

Q2-1. The construct of B.1.617.1 as specified in Fig. S1A differs from the commonly used definition of the B.1.617.1 (<https://www.cdc.gov/coronavirus/2019-ncov/variants/variant-info.html>) in that G142D and Q1071H are missing in this study. The authors ought to clarify why these mutations are not included in the construct, and how these differences may impact on the structure and function of the spike protein.

A2-1: Thanks for pointing this out. We started our study on the B.1.617.1 (Kappa) variant in the early spring of this year and the plasmids were constructed based on an early sequence of Kappa (hCoV-19/India/WB-1931500939910/2021, GISAID ID:

EPI_ISL_1589917) which does not have the G142D and Q1071H mutations. We should mention that, at that time, the CDC has not defined the Kappa variant sequence (<https://www.cdc.gov/coronavirus/2019-ncov/variants/variant-info.html>). As for the structural impacts of these two mutations, so far, there is no available structure of Kappa or other S structures containing these two mutations in the PDB databank. Although a recent BioRxiv manuscript³ reports the structure of Kappa variant that contains the G142D and Q1071H mutations, still the structural impact of these two mutations remains unknown. Based on our current Kappa S-open structure, the Q1071H mutation, located in S2 subunit, does not change the original contacts with the P715-N717 region (within 4 Å distance, Fig. R2), and hence may not affect the overall conformation of the spike, especially for the upper S1 subunit. The G142D mutation locates in the supersite of NTD (Fig. R2A), and it has been previously shown through biochemical study that G142D abrogates binding of some NTD-specific neutralizing mAbs⁴, therefore we postulate that the G142D in Kappa may not significantly impact the overall spike structure, rather, it may contribute to the immune evasion of Kappa..

Fig. R2 The G142D and Q1071H mutations in Kappa variant. (A) The locations of G142D and Q1071H mutations in Kappa protomer 1 based on our current Kappa S-open structure. (B) Zoomed-in view of the Q1071H mutation site, showing that the Q1071H mutation does not change the original contacts with the P715-N717 region (within 4 Å distance).

Q2-2. The authors used BLI analyses and WT-S as a reference to compare the ACE2 binding affinities of individual spike variants. Fig. 1A showed a three-fold reduction in

ACE2 binding affinity for G614-S, a 50% reduction for B.1.617.1-S and a two-fold increase for B.1.351-S. First of all, the authors only presented the raw sensor grams without showing the fitting results, making it difficult to evaluate the quality of the experimental data. Some flaw in the data quality is already evident as the dissociation curves often overlap (WT-S and G614G-S) and the spacing between different sensor grams are not appropriately distributed according to the specified protein concentrations. Furthermore, the authors did not specify the construct design of ACE2 in the Method section in which the expression and purification of recombinant spike variants and ACE2 are described. These technical problems raise the issue of the validity of the BLI analysis.

A2-2: Thanks for the comment. To obtain more objective and credible measurement results and improve the data quality, we have redone the biotin labeling experiments of S proteins and BLI experiments. Briefly, 88 μg of individual prefusion-stabilized S trimer was incubated with 4.5 μL of Sulfo-NHS-LC-LC-Biotin (10 mM) on ice for 2.5 h, followed by desalting. The streptavidin (SA) biosensors were loaded with the biotinylated S trimer and then dipped into wells containing varying concentrations of monomeric ACE2. As shown in Fig R3, the new data (including the raw sensor grams and the fitting results) were of high quality, thus allowing accurate and reliable comparison of ACE2-binding affinity between the different S trimers analyzed.

The new dataset shows a similar trend of ACE2 binding affinity change for the variants as in the previous dataset: G614 S exhibited lower affinity than that of Beta (B.1.351), Kappa (B.1.617.1), or WT. Specifically, ACE2 binding affinities of WT-S, G614-S, Kappa-S, and Beta-S were determined to be 104, 365, 84, and 83 nM, respectively. The new data have been added in our revised manuscript (Fig. 1A) and also shown below for the convenience of the reviewers and editor (Fig R3).

Fig. R3 Measurement of binding affinity between S trimers and ACE2 using bio-layer

interferometry (BLI). Association and dissociation steps are divided by dotted lines. ACE2 concentrations tested were shown. Raw sensor gram curves and fitting curves were shown in color and black, respectively.

The construct design of ACE2 has been described in our previous work², and therefore we did not introduce it in detail in the previous version of the manuscript. We have now specified the construct design of ACE2 in the Method section of our revised manuscript, to read “A gene encoding human ACE2 PD domain (Q18-D615) with an N-terminal interleukin-10 (IL-10) signal peptide and a C-terminal His tag was cloned into vector pcDNA 3.4.” (please see Page 13, Line 372-374).

Q2-3. Tokunaga and co-workers recently reported an increase of ACE2 binding for the D614G variant, which contrasts the opposite results as described herein. Using the same BLI analysis, their K_d values are two orders of magnitude lower than the values reported in this study. The difference in the absolute scales and the relative affinity between these two studies should be addressed.

A2-3: After receiving the review comments, we have collected information on the ACE2-binding affinity of S trimer (WT and variants)^{1,5,6} and summarized them in Table R1. As shown in Table R1, the K_D values determined by different groups varied, likely due to multiple factors, including the methods used (BLI or SPR), S protein used (full-length S or prefusion-stabilized S ectodomain), ACE2 used (monomeric or dimeric), protein loaded onto sensor (ACE2 or S), and even binding temperature. In our current study, to be consistent with our structural analysis, we used monomeric ACE2 (human ACE2 PD domain, please see A2-2) and stabilized S ectodomain for BLI analyses. Specifically, streptavidin (SA) biosensors were loaded with the biotinylated prefusion-stabilized S ectodomain (S2P) and then dipped into wells containing different concentrations of ACE2 protein. Our new BLI results showed that the K_D values for WT, G614, Kappa, Beta S were 104, 365, 84, and 83 nM, respectively (Fig. R3). Our results are in general consistent with the data reported previously (Table R1). For example, the BLI data (generated with full-length S) from Bing Chen lab showed that G614 S bound ACE2 monomer less tightly than did the WT S¹ and the Beta S⁷. In addition, a recent preprint reported that the full-length S of G614 had lower affinity to ACE2

monomer than that of Kappa ($K_D=288$ vs 160 nM)⁶.

In the study by Ozono et al. from Tokunaga lab⁶, the BLI assays were performed in a way different from ours. Specifically, in that study, the AHC biosensors were loaded with ACE2 dimerized by a Fc tag and then dipped into wells containing stabilized ectodomain of WT S or D614G S trimer. Their BLI data showed that the D614G S exhibited markedly higher (>14.6 fold) affinity to ACE2 dimer relative to WT S at 25 degrees, however, the increase in ACE2 dimer-binding affinity for D614G relative WT S was only modest (2.1-fold) or marginal (1.5-fold) at 30 or 37 degrees, respectively⁶ (Table R1). We should emphasize that ACE2 dimer (linked by Fc) was used as the source of ACE2 in that study. It has been reported that, for a given S, its affinity to ACE2 dimer is usually much higher than that to ACE2 monomer^{1,7,8} (Table R1). Hence, we reason that the difference in the affinity between Ozono et al.⁶ and our studies is likely caused by the ACE2 protein used (monomer vs dimer) and the biosensor loading method (with ACE2 or S).

Collectively, our BLI data are in general consistent with most of the S/ACE2 affinity results reported thus far (Table R1), despite seemingly contradictory to those from Tokunaga lab⁶. As suggested by the reviewers, we have now described these data consistency and discussed possible causes of the discrepancy in our revised manuscript (please see Page 4, Line 90-93, 96-98).

Table R1 Binding affinity of S trimer (WT and the variants) to ACE2 reported recently

Reference	SARS-CoV-2 strain	Method	Sensor	Loading protein	Analyte	Temperature (°C)	K _D (nM)
This study	WT, G614, Kappa, and Beta	BLI	SA biosensors	Prefusion-stabilized S ectodomain	Monomeric ACE2	25	WT: 104.10
							G614: 365.00
							Kappa: 84.22
							Beta: 82.66
J. Zhang et al. , Science , 10.1126/science.abf2303 ⁷	WT and G614	BLI	AR2G biosensors	Full-length S protein	Dimeric ACE2		WT: 11.2
					Monomeric ACE2		G614: 15.5
						WT: 133.0	
						G614: 343.0	
L. Yurkovetskiy et al. , Cell ,	WT and G614	SPR	Series S CM5	ACE2.Fc	Soluble spike	25	WT: 1.38
							G614: 7.97

10.1016/j.cell.2020.09.032. ⁵			sensor chip		trimer protein	37	WT: 0.71 G614: 3.76
S. Ozono et al. , Nat Commun , 10.1038/s41467-021-21118-2. ⁶	WT and G614	BLI	AHC biosensor	Fc-tagged ACE2-ectodomain dimer	Stabilized S ectodomain	25	WT: 0.0146 G614: < 0.001
						30	WT: 0.614 G614: 0.297
						37	WT: 1.58 G614: 1.04
Y. Cai et al. , Science , 10.1126/science.abi9745. ⁷	G614, Alpha, and Beta	BLI	AR2G biosensors	Full-length S protein	Dimeric ACE2		G614: 12.8 Alpha: 2.7 Beta: 15.1
					Monomeric ACE2		G614: 124.0 Alpha: 45.9 Beta: 71.4
S. M.-C. Gobeil et al. , Science , 10.1126/science.abi6226. ⁹	G614, Alpha, and Beta	SPR	Series S Streptavidin (SA) chip	Stabilized S ectodomain	ACE2-Fc		G614: 218.29 Alpha: 45.20 Beta: 72.63
J. Zhang et al. , BioRxiv , 10.1101/2021.08.17.456689. ⁸	G614, Gamma, Kappa, and Delta	BLI	AR2G biosensors	Full-length S protein	Dimeric ACE2		G614: 19.6 Gamma: 2.4 Kappa: 7.5 Delta: 176
					Monomeric ACE2		G614: 288.0 Gamma: 14.8 Kappa: 160.0 Delta: 208.0
M. McCallum et al. , BioRxiv , 10.1101/2021.08.11.455956. ³	Alpha, Kappa, Delta and Delta+	BLI	SA biosensors	His-avi-tagged RBD	Monomeric ACE2		WT: 147 Alpha: 26 Kappa: 88 Delta: 180 Delta+: 550

Q2-4. Likewise, the BLI analyses reported by the two recent Science papers (10.1126/science.abf2303 & 10.1126/science.abi6226) on the B.1.351 variants and the two preprints (<https://doi.org/10.1101/2021.08.11.455956> and <https://www.biorxiv.org/content/10.1101/2021.08.17.456689v1>) on the Kappa variant should be commented. In particular, the Kappa variant was found to bind to ACE2 much more strongly than WT in both preprints, but the current study showed otherwise. The potential source of the contraction should be appropriately addressed.

A2-4: Thanks for the constructive comment. The affinity data reported in the papers mentioned by the reviewer are now listed in the above Table R1, along with our new BLI results for comparison.

Specifically, the Science paper ([10.1126/science.abf2303](https://doi.org/10.1126/science.abf2303))¹ from Bing Chen lab reported that the G614 S bound monomeric ACE2 less tightly ($K_D=343$ nM) than did the WT S ($K_D=133$ nM) as determined by BLI assays. These results are in well agreement with our BLI data ($K_D= 365$ nM for G614 S and 104 nM for WT S towards monomeric ACE2).

The Science paper ([10.1126/science.abi6226](https://doi.org/10.1126/science.abi6226))⁹ mentioned by the reviewer is from Priyamvada Acharya lab. In this work, the authors measured spike binding to the ACE2 receptor ectodomain (Fc tagged) by using surface plasmon resonance (SPR) and ELISA. Their SPR results showed that Beta S bound stronger ($K_D=72$ nM) to ACE2 than did the G614 S ($K_D=218$ nM). In addition, another Science paper ([10.1126/science.abi9745](https://doi.org/10.1126/science.abi9745))⁷ from Bing Chen lab also reported that Beta S has higher affinity ($K_D=71.4$ nM) to monomeric ACE2 than G614 S ($K_D=124$ nM) as measured by BLI. The results of these two Science papers are in line with our BLI data ($K_D= 365$ nM for G614 S and 83 nM for Beta S towards monomeric ACE2).

The preprint (<https://doi.org/10.1101/2021.08.11.455956>)³ mentioned by the reviewer is from David Veessler and co-workers. The authors analyzed the ACE2 binding of RBDs from WT or Kappa variant using ELISA, SPR, and BLI. Their BLI results showed that Kappa RBD binds to monomeric ACE2 more strongly ($K_D=88$ nM) than does WT RBD ($K_D=147$ nM). Our new BLI data also showed that Kappa S trimer bound slightly stronger to ACE2 monomer than did the WT S ($K_D= 84$ nM for Kappa S and 104 nM for WT S towards monomeric ACE2), in line with the result from David Veessler lab³.

The preprint (<https://www.biorxiv.org/content/10.1101/2021.08.17.456689v1>)⁸ is from Bing Chen lab. The authors reported that the full length S of Kappa (B.1.617.1) bound to monomeric ACE2 more strongly ($K_D=160$ nM) than did G614 S ($K_D=288$ nM), as determined by BLI. These results are consistent with our new BLI data ($K_D= 365$ nM for G614 S and 84 nM for Kappa S towards monomeric ACE2).

Collectively, our BLI data are consistent with the S/ACE2 affinity results for Beta/Kappa variants reported thus far (Table R1). We have now followed the suggestion from our

reviewer to describe these data consistency in our revised manuscript (please see Page 4, Line 93-98).

Q2-5. With regard to the geometrical analysis of the RBD conformational changes as illustrated in Fig. 2D/F/H, Fig. 3D/E/G, and Fig. 6, the points of rotation (hinge) should be explicitly defined as the portions of the hinge can make a big difference in the results of the rotation angles. In particular, it is unclear whether the RBDs in Fig. 2H and Fig. 3G are rotating around the long axis defined by the central helix of the S2 subunit of out of plane whose normal is parallel to the long axis of the central helix, i.e., an upward motion as defined in Fig. 2D. The repeating 3DVA analysis results of Fig. 2G-H and Fig. 3F-G are somewhat superfluous since they report the same features.

A2-5: Thanks for pointing this out. For the rotation angle measurement, we used the 'measure rotation' command in UCSF Chimera to automatically calculate the rotation of interested structural elements between two states (Fig. 2F, Fig. 3D) or between two extremes in 3DVA motions (Fig. 2H, Fig. 3G, Fig. 6). For the "up" RBD-1 with or without associated ACE2 in the S trimer for both variants, its rotation is around the lower part of SD1 (Fig. R4A and Fig. S5C); for the originally "down" RBD-2/RBD-3, their tilting up is also around the SD1 (Fig. R4A and Fig. S5C). The NTD rotation is around the region spatially between NTD and SD2 (near the flexible 304-310 loop) (Fig. R4B, Fig. 2F, 3D and Fig. S2G). Similar rotation hinges for these key structural elements in S trimers have also been observed in other reports^{9,10}. Collectively, the RBDs and NTDs rotate around its individual hinges at distinct structural motifs, instead of the pseudo C3 rotational axis of the S trimer to some extent parallel to the S2 central helix. We have now added the rotational angle measurement in the Method session (Page 17, Line 502), and related description and figures (Fig. S2G and S5C) about the rotation hinges related to Fig. 2D/F/H, Fig. 3D/E/G, and Fig. 6 in our revised manuscript.

For the free Kappa S trimer, our 3DVA mainly displays a "breath" motion (Movie S1), with the three NTD-RBD pairs (including NTD-1/RBD-2, NTD-2/RBD-3 and NTD-3/RBD-1) tilting outward/downward simultaneously, consequently the entire S trimer appears untwisted and expanded (Movie S1, Fig. 2G-H). While for the free Beta S trimer, the S1 subunit RBD-3 movement could be allosterically transferred to the central S2 helix bundle through contacting the HR1-CH hairpin, accordingly S2 could

also be involved in the allosteric cooperation with S1 movement (Movie S2, Fig. 3F-G, Fig. S3G), not seen in Kappa counterpart. We have now modified the related text (Page 7, Line 179-192) and Movies S1-S2 to make this clearer.

Fig. R4 Analysis of the rotation of key motifs. (A) The 3DVA mode 1 motion of Kappa S-ACE2 displayed in maps, showing two extreme conditions in motion, with the rotation hinges for RBD-1 (left) and RBD-3 (right) indicated by blue stars. (B) Protomer 1 from the overlaid structures of Kappa S-open (in color) and G614 (PDB: 7KRR, purple), and zoomed-in view of the NTD-SD2 region, with the NTD rotation hinge, located around the region spatially between NTD and SD2, indicated by a blue star.

Q2-6. The functional implications of the RBD motions should be better elaborated. For example, in line 208, why would more open spike be beneficial for the shedding of S1? Appropriate reference should be included.

A2-6: Thanks for the suggestion. More dynamic RBD motion is beneficial for RBDs' upward tilting to the "up" position and eventually more open spike. For SARS-CoV-2 S protein, RBD in the "up" position indicates the receptor-accessible state^{2,11,12}. It has also been suggested that RBD in the up position reduces the interaction between S1 and S2, and specifically releases the constraints imposed on the HR1-CH hairpin, which is known to completely refold during the membrane fusion process^{13,14}. Thus, the more open spike (with more "up" RBDs) would be beneficial for the transformation of S trimer towards the postfusion state and the simultaneous shedding of S1^{2,15-18}. These are the functional implications of the RBD motions, which have been included in our revised manuscript (line 210-214 on Page 8). In line 208, "more open spike"

means the Kappa S trimer tends to be in the more open C2a/C2b (51.7% population, two RBDs up) and C3 (34.1%, all three RBDs up) states with two or three RBDs in the “up” configuration. We have also supplied corresponding references in the revised manuscript.

Q2-7. Discussion, first paragraph, “our 3DVA data suggested pronounced conformational dynamics especially for B.1.351 S trimer with S2 potentially involved in the allosteric cooperation in S1 movement.” How can the 3DVA analysis inform us on allostery? More specifically, it not very clear from Fig. 2G-H and 3F-G what allostery can be inferred from the motions. The authors should define the allostery in the context of spike dynamics with better schematic representations.

A2-7: Our 3DVA data show that different portions of the S trimer undergo continuous motions in a coordinated way, thereby revealing the allosteric network of the S trimer. This is the unique power of 3DVA. Indeed, 3DVA has been adopted to analyze other complexes, revealing the coordinated motion and the underlying allostery network of the complexes^{19,20}. For the Beta S trimer, as showed in Fig. R5, Fig. 3F-G and Movie S2, our 3DVA data suggested that, with the down/up movement, RBD-3 (belonging to S1 subunit) contacts/leaves the S2 HR1-CH hairpin from the neighboring protomer 2 (Fig. R5), which could be propagated to the S2 helix bundle, inducing its down/up movement. This S2 helix bundle movement could even be propagated to the membrane-proximal stalk. Collectively, for Beta S trimer, the S1 subunit RBD-3 movement could be allosterically transferred to the central S2 helix bundle through contacting the HR1-CH hairpin, accordingly S2 could also be involved in the allosteric cooperation with S1 movement.

As for the allostery in Fig. 2G-H for the Kappa S trimer, our 3DVA suggested that the neighboring NTD and RBD can form three NTD-RBD pairs, including NTD-1/RBD-2, NTD-2/RBD-3, and NTD-3/RBD-1. The three NTD-RBD pairs tilt outward/downward simultaneously, consequently the entire S trimer appears untwisted and expanded (Movie S1, Fig. 2G-H), which could release the protomer interaction strength, beneficial for the transient raising up of RBD. We have now modified related contents to clarify the allosteric network revealed by our 3DVA, indicated the related allosteric

structural elements in Fig. 2G-H and 3F-G and Movies S1-S5, and included Fig. R5 as Fig. S5G in the revised manuscript.

Fig. R5 The 3DVA motion of Beta S-open, displayed in two extreme maps (in royal blue and grey, respectively) in the motion. The S1 subunit RBD-3 movement could be allosterically propagated to the central S2 helix bundle of the neighboring protomer 2 through contacting the HR1-CH hairpin.

Minor comments:

1. The sentence in line 192 (Four cryo-EM map,...) is very long and hard to read. It should be rephrased.

A: This sentence has been rephrased, to read “Four cryo-EM maps of the Kappa S trimer engaged with ACE2, including Kappa S-ACE2-C1 (only RBD-1 up), S-ACE2-C2a (RBD-1 and RBD-2 up), S-ACE2-C2b (RBD-1 and RBD-3 up), and S-ACE2-C3 (all three RBDs up), were determined at 4.0, 3.9, 3.9, and 3.9-Å-resolution, respectively (Fig. 4A-D, S4A-C, Table S1).”

2. The dissociation constant K_d should be subscript (Fig. 1A) and the on- and off-rates, k_{on} and k_{off} , should be lower case k with on and off (dis is an unusual usage for the off rate) as subscripts.

The suggestion is well taken. The writing of the dissociation constant in Fig.1A was changed in our revised manuscript.

3. IC₅₀ in Fig. 1C (50 should be subscript, too)

We have changed it to IC_{50} with 50 be subscript.

4. Fig. S6. It may be the threshold issue that caused the loss of the ACE2 EM density in Fig. S6E C2 and C2b when three and two ACE2 are expected but only one is shown. This is contrasting the result in Fig. S6F in which three ACE2 are shown although two show weaker density.

A: In Fig. S6E, we intended to display the local resolution of the maps with reasonable high-resolution structural features, but not to show the extra bound ACE2 density. It is indeed the threshold issue that only one ACE2 was shown for C2a and C2b in Fig. S6E. Actually, when we lower the threshold, the C2a/C2b maps can also display some density of an additional bound ACE2 in addition to the static one engaged with RBD-1 (please see Fig.R6). We have now added the lower threshold rendering of the C2a/C2b maps as an extra panel in Fig. S4F and S6F.

Fig. R6 Lower threshold rendering of the Beta S-ACE2-C2a/C2b map, showing ACE2 density also associated with RBD-2 or RBD-3 in the C2a and C2b map, respectively.

References:

- 1 Zhang, J. *et al.* Structural impact on SARS-CoV-2 spike protein by D614G substitution. *Science* **372**, 525-530, doi:10.1126/science.abf2303 (2021).
- 2 Xu, C. *et al.* Conformational dynamics of SARS-CoV-2 trimeric spike glycoprotein in complex with receptor ACE2 revealed by cryo-EM. *Sci Adv* **7**, doi:10.1126/sciadv.abe5575 (2021).
- 3 McCallum, M. *et al.* Molecular basis of immune evasion by the delta and kappa SARS-CoV-2 variants. *bioRxiv*, doi:10.1101/2021.08.11.455956 (2021).
- 4 McCallum, M. *et al.* N-terminal domain antigenic mapping reveals a site of vulnerability for SARS-CoV-2. *Cell* **184**, 2332-2347 e2316, doi:10.1016/j.cell.2021.03.028 (2021).
- 5 Yurkovetskiy, L. *et al.* Structural and Functional Analysis of the D614G SARS-CoV-2 Spike Protein Variant. *Cell* **183**, 739-751 e738, doi:10.1016/j.cell.2020.09.032 (2020).
- 6 Ozono, S. *et al.* SARS-CoV-2 D614G spike mutation increases entry efficiency with enhanced ACE2-binding affinity. *Nat Commun* **12**, 848, doi:10.1038/s41467-021-21118-2 (2021).
- 7 Cai, Y. *et al.* Structural basis for enhanced infectivity and immune evasion of SARS-CoV-2 variants. *Science*, doi:10.1126/science.abi9745 (2021).
- 8 Zhang, J. *et al.* Membrane fusion and immune evasion by the spike protein of SARS-CoV-2 Delta variant. *bioRxiv*, doi:10.1101/2021.08.17.456689 (2021).
- 9 Gobeil, S. M. *et al.* Effect of natural mutations of SARS-CoV-2 on spike structure, conformation, and antigenicity. *Science*, doi:10.1126/science.abi6226 (2021).
- 10 Yang, T. J. *et al.* Effect of SARS-CoV-2 B.1.1.7 mutations on spike protein structure and function. *Nature structural & molecular biology* **28**, 731-739, doi:10.1038/s41594-021-00652-z (2021).
- 11 Wrapp, D. *et al.* Cryo-EM structure of the 2019-nCoV spike in the prefusion conformation. *Science* **367**, 1260-1263, doi:10.1126/science.abb2507 (2020).
- 12 Walls, A. C. *et al.* Structure, Function, and Antigenicity of the SARS-CoV-2 Spike Glycoprotein. *Cell* **181**, 281-292 e286, doi:10.1016/j.cell.2020.02.058 (2020).

-
- 13 Walls, A. C. *et al.* Tectonic conformational changes of a coronavirus spike glycoprotein promote membrane fusion. *Proc Natl Acad Sci U S A* **114**, 11157-11162, doi:10.1073/pnas.1708727114 (2017).
 - 14 Walls, A. C. *et al.* Unexpected Receptor Functional Mimicry Elucidates Activation of Coronavirus Fusion. *Cell* **183**, 1732, doi:10.1016/j.cell.2020.11.031 (2020).
 - 15 Benton, D. J. *et al.* Receptor binding and priming of the spike protein of SARS-CoV-2 for membrane fusion. *Nature* **588**, 327-330, doi:10.1038/s41586-020-2772-0 (2020).
 - 16 Zhou, T. *et al.* Cryo-EM Structures of SARS-CoV-2 Spike without and with ACE2 Reveal a pH-Dependent Switch to Mediate Endosomal Positioning of Receptor-Binding Domains. *Cell Host Microbe* **28**, 867-879 e865, doi:10.1016/j.chom.2020.11.004 (2020).
 - 17 Yan, R. *et al.* Structural basis for the different states of the spike protein of SARS-CoV-2 in complex with ACE2. *Cell research* **31**, 717-719, doi:10.1038/s41422-021-00490-0 (2021).
 - 18 Zhu, X. *et al.* Cryo-electron microscopy structures of the N501Y SARS-CoV-2 spike protein in complex with ACE2 and 2 potent neutralizing antibodies. *PLoS Biol* **19**, e3001237, doi:10.1371/journal.pbio.3001237 (2021).
 - 19 Punjani, A. & Fleet, D. J. 3D variability analysis: Resolving continuous flexibility and discrete heterogeneity from single particle cryo-EM. *J Struct Biol* **213**, 107702, doi:10.1016/j.jsb.2021.107702 (2021).
 - 20 Kudryavtseva, S. S. *et al.* Novel cryo-EM structure of an ADP-bound GroEL-GroES complex. *Sci Rep* **11**, 18241, doi:10.1038/s41598-021-97657-x (2021).

REVIEWER COMMENTS

Reviewer #1 (Remarks to the Author):

The authors have addressed the points raised previously. The current manuscript is suitable for acceptance.

Reviewer #2 (Remarks to the Author):

The authors made significant efforts in the revised manuscript to address the reviewers' comments. However, some issues remain to be addressed to ensure the reproducibility and consistency of the reported findings.

Major comments:

1. The authors' reply on the lack of the G142D and Q1071H mutations in their Kappa variant is well taken, but the difference from the WHO's definition should be explicitly stated in the manuscript even if the G142D and Q1071H mutations do not appear to make significant contributions to ACE2 binding or RBD conformational changes. This is an important factor to consider when the structures will be used by readers who may overlook the subtle but potentially important differences.
2. I am of the opinion that the cryo-EM data themselves do not provide information with regard to allostery unless some functional readouts can be derived from the conformational changes. The additional EM density corresponding to the C-terminal helix bundle cannot be simply ascribed to allosteric regulation mediated by the HR1-CH hairpin (Fig. R5; Fig. S3G). The lack of well-defined EM density in dynamic regions can be attributed to multiple factors in addition to allostery. I would suggest to change the term "allostery" to "collective motions" in the 3DVA analyses presented in Fig. 2G-H and 3F-G.
3. With regard to the BLI kinetic analyses of ACE2 binding to the four S variants, a number of studies related to this study have been reported during the reviewing process of this manuscript. These include full-length spike variants of Beta, Gamma, Delta and Kappa <https://doi.org/10.1101/2021.09.12.459978> & <https://doi.org/10.1101/2021.09.02.458774>. RBD of WT and Delta <https://doi.org/10.1101/2021.09.03.458829>. These studies (especially the first two) should be discussed in the manuscript since they also studied the Beta and Kappa variants.
4. The reported dissociate constants (K_d) in the above studies are much lower than the ones reported in the current study. Importantly, the BLI-derived response units in the above studies (full ectodomain of S variants) are significantly higher than the revised results reported herein, which showed improvements in the dose-dependent kinetic traces but the absolute values remain problematic. The results reported herein is about 10 times lower than the other studies. It is likely that the immobilization of the S variants by biotinylation in the current study is affecting the conformations of the RBDs, or generating too much steric hinderance to ACE2 binding, such that the amounts of ACE2 binding-competent S variants are much lower than expected. If so, this will also affect the K_d values derived from the global fitting.
5. In particular, it is unclear why the authors used a different concentration range for G614-S: maximum ACE2 concentration of 1800 nM instead of 200 nM as used in the other three variants (Fig. 1A). Such a concentration is unusually high for ACE2 binding analyses compared to the literature. The authors should keep the data presentation consistent unless there is a clear reason to make specific changes to only one of the four S variants.
6. With regard to the definition of conformational changes in the S protein structures, the authors replied that the command 'measure rotation' in Chimera was used to calculate the rotation angles. The output of the function contains more informational than the single rotation angle. According to the manual of Chimera, the function will also report the transformation of model2 relative to model1 as:
* a matrix in which the first three columns describe a rotation and the fourth describes a translation (performed after the rotation)
* an axis of rotation (a unit vector), point on the axis, rotation angle, and shift parallel to the axis
The latter parameter, namely the axis of rotation, is important as it will define how the rotation angle is

calculated with respect to which axis. Such information is necessary for others to reproduce the calculations in future studies. It is particularly important when the RBD/NTD motions are described by different geometrical definitions, and comparisons between this study and their studies need to be made.

Minor comments:

1. The choice of colors in Fig. 2F, Fig. 3D and Fig. 3E should be improved as some of the colors are very similar, making it challenging to visually compare the differences.

REVIEWER COMMENTS

Reviewer #1

The authors have addressed the points raised previously. The current manuscript is suitable for acceptance.

--Thanks for the positive comments and insightful suggestions from this reviewer.

Reviewer #2

The authors made significant efforts in the revised manuscript to address the reviewers' comments. However, some issues remain to be addressed to ensure the reproducibility and consistency of the reported findings.

--We appreciate all the encouraging comments and suggestions from this reviewer, which led us to explore more thoroughly to improve our manuscript.

Major comments:

Q2-1. The authors' reply on the lack of the G142D and Q1071H mutations in their Kappa variant is well taken, but the difference from the WHO's definition should be explicitly stated in the manuscript even if the G142D and Q1071H mutations do not appear to make significant contributions to ACE2 binding or RBD conformational changes. This is an important factor to consider when the structures will be used by readers who may overlook the subtle but potentially important differences.

A2-1: The suggestion is well taken. We have now included the related information in our revised manuscript in Method (Page 14, Line 392-395), to read "Note that the sequence of the Kappa variant adopted here is derived from an earlier Kappa strain and hence does not contain the G142D and Q1071H mutations compared to the Kappa variant sequence defined later by WHO (<https://www.cdc.gov/coronavirus/2019-ncov/variants/variant-info.html>)".

Q2-2. I am of the opinion that the cryo-EM data themselves do not provide information with regard to allostery unless some functional readouts can be derived from the conformational changes. The additional EM density corresponding to the C-terminal helix bundle cannot be simply ascribed to allosteric regulation mediated by the HR1-CH hairpin (Fig. R5; Fig. S3G). The lack of well-defined EM density in dynamic regions can be attributed to multiple factors in addition to allostery. I would suggest to

change the term “allostery” to “collective motions” in the 3DVA analyses presented in Fig. 2G-H and 3F-G.

A2-2: We have followed the suggestion to replace “allostery” to “collective motions” in the 3DVA analyses in our revised manuscript (please see Line 150 on P. 6 and Line 188 on P. 7).

Q2-3. With regard to the BLI kinetic analyses of ACE2 binding to the four S variants, a number of studies related to this study have been reported during the reviewing process of this manuscript. These include full-length spike variants of Beta, Gamma, Delta and Kappa <https://doi.org/10.1101/2021.09.12.459978> & <https://doi.org/10.1101/2021.09.02.458774>. RBD of WT and Delta <https://doi.org/10.1101/2021.09.03.458829>. These studies (especially the first two) should be discussed in the manuscript since they also studied the Beta and Kappa variants.

A2-3: Thanks for pointing out these new BioRxiv preprints. We have now discussed the first two studies^{1,2} (<https://doi.org/10.1101/2021.09.12.459978> & <https://doi.org/10.1101/2021.09.02.458774>) in our revised manuscript (please see page 11, lines 303-308). To read “It should be mentioned that, for a given variant S protein, its absolute K_D values generated from different studies, including two studies recently reported on BioRxiv, may vary significantly, likely due to multiple factors such as the method used (BLI or SPR), S protein construct (full-length S or prefusion-stabilized S ectodomain), ACE2 form (monomeric or dimeric), protein used to load the sensors (ACE2 or S), and even binding temperature.”

The third preprint³ (<https://doi.org/10.1101/2021.09.03.458829>) mentioned by this reviewer actually reported the binding kinetics of ACE2 proteins (human or mouse) with the RBD of WT or Delta variant determined by SPR. We elected not to discuss this work because our current study does not involve Delta.

Q2-4. The reported dissociate constants (K_d) in the above studies are much lower than the ones reported in the current study. Importantly, the BLI-derived response units in the above studies (full ectodomain of S variants) are significantly higher than the revised results reported herein, which showed improvements in the dose-dependent kinetic traces but the absolute values remain problematic. The results reported herein is about 10 times lower than the other studies. It is likely that

the immobilization of the S variants by biotinylation in the current study is affecting the conformations of the RBDs, or generating too much steric hinderance to ACE2 binding, such that the amounts of ACE2 binding-competent S variants are much lower than expected. If so, this will also affect the K_D values derived from the global fitting.

A2-4: We understand this reviewer's concern. As mentioned in our last responding letter, the K_D values determined by different groups varied, likely due to multiple factors, including the methods used (BLI or SPR), S protein used (full-length S or prefusion-stabilized S ectodomain), ACE2 used (monomeric or dimeric), protein loaded onto sensor (ACE2 or S), and even binding temperature (please see Table R1 below). It has been reported that, for a given S, its affinity to ACE2 dimer is much higher than that to ACE2 monomer (about 10 times, Table R1), and the response units are also much higher⁴⁻⁶. Specifically, Cai *et al.*⁵, (*Science*, 10.1126/science.abi9745.) showed that the full-length G614 S had a K_D value of 12.8 nM to dimeric ACE2 but its K_D value to monomeric ACE2 was 124 nM, highlighting the drastic influence of protein construct and BLI protocol in absolute K_D values obtained.

In our study, we performed BLI to determine the affinity of variant S ectodomain to monomeric ACE2. Specifically, streptavidin (SA) biosensors were loaded with the biotinylated prefusion stabilized S ectodomain (S2P) and then dipped into wells containing different concentrations of monomeric ACE2 protein. Our BLI results showed that the K_D values for WT, G614, Kappa, Beta S were 104, 365, 84, and 83 nM, respectively. Our data are in general consistent (both in the scale of absolute values and in the trend of affinity change among variants) with those generated with a similar protocol⁴⁻⁶ (Table R1). For example, Zhang *et al.*⁶ (*Science*, 10.1126/science.abf2303) reported that the K_D values of full-length WT and G614 S to monomeric ACE2 were 133 nM and 343 nM, respectively. Regarding the concern whether the immobilization of the S variants by biotinylation in our study would affect the absolute K_D values, we do not have a definite answer to that; however, the data from other studies suggest that biotinylation has no or minimal impact on the affinity values. Specifically, the K_D values of biotinylated Kappa S for dimeric ACE2⁴ (Zhang *et al.*, BioRxiv, 10.1101/2021.08.17.456689.) were comparable ($K_D=7.5$ vs 4.88 nM) to that of non-biotinylated Kappa S to dimeric ACE2 loaded onto the Protein-A biosensors² (Saville *et al.*, BioRxiv, <https://doi.org/10.1101/2021.09.02.458774>),

indicating that biotinylation of S does not significantly affect the conformations of the RBDs. Also, we should mention that the BLI protocol or the similar ones have been validated in our and others' published works⁴⁻⁷.

We have carefully read the first two preprints mentioned by the reviewer and found that the protocols/methods used in these studies are different from ours. Specifically, in the first study¹ (<https://doi.org/10.1101/2021.09.12.459978>), the sensors were loaded with biotinylated sfGFP-ACE2 and then detected for S binding, and the determined K_D values for Beta, Gamma, Delta, and Kappa S proteins were 0.25, 0.30, 0.39, and 0.09 nM, respectively. In the second study² (<https://doi.org/10.1101/2021.09.02.458774>), the sensors were loaded with (dimeric) ACE2-mFc and then measured for S binding, and the K_D values of D614G, Delta, and Kappa S proteins were determined to be 5.06, 2.65, and 4.88 nM, respectively. These results are actually quite surprising (eg. for the Kappa S, the K_D value reported in the first study is more than 50 times lower (0.09 nM vs 4.88 nM) than that in the second study), because one would expect higher affinity of S to dimeric ACE2-mFc than to monomeric sfGFP-ACE2 as seen in other published works⁴⁻⁶. Again, this inconsistency is likely due to the variation in the construct/protein design and experiment protocols used by different groups. We have now discussed this aspect in the revised manuscript (Page11, lines 303–312). Nonetheless, we feel that, when the affinity data generated by different groups are analyzed, it might be more meaningful to look at the trend of affinity changes for different S variants rather than to compare individual absolute affinity values between studies.

Table R1 Binding affinity of S trimer (WT and the variants) to ACE2 reported recently

Reference	SARS-CoV-2 strain	Method	Sensor	Loading protein	Analyte	Temperature (°C)	K_D (nM)
This study	WT, G614, Kappa, and Beta	BLI	SA biosensors	Prefusion-stabilized S ectodomain	Monomeric ACE2	25	WT: 104.10
							G614: 365.00
							Kappa: 84.22
							Beta: 82.66
T. Yang et al. , BioRxiv, https://doi.org/10.1101/2021.09.12.459978 ¹	Beta, Gamma, Delta, and Kappa,	BLI	SAX biosensors	sfGFP-ACE2	Stabilized S ectodomain	25	Beta 0.25
							Gamma: 0.3
							Delta: 0.39
							Kappa: 0.09
J. Saville et al. , BioRxiv,	G614, Kappa,	BLI	Protein-A biosensors	ACE2-mFc	Prefusion-stabilized S		G614: 5.06
							Kappa: 4.88

https://doi.org/10.1101/2021.09.02.458774 ²	Delta				ectodomain			Delta: 2.65
W. Ren et al. , BioRxiv , https://doi.org/10.1101/2021.09.03.458829 ³	WT and Delta	SPR	CM5 chip	Monomeric ACE2	RBD			WT: 5.5 Delta: 2.67
J. Zhang et al. , Science , 10.1126/science.abf2303 ⁶	WT and G614	BLI	AR2G biosensors	Full-length S protein	Dimeric ACE2			WT: 11.2 G614: 15.5
Y. Cai et al. , Science , 10.1126/science.abi9745. ⁵	G614, Alpha, and Beta	BLI	AR2G biosensors	Full-length S protein	Dimeric ACE2			WT: 133.0 G614: 343.0
					Monomeric ACE2			G614: 12.8 Alpha: 2.7 Beta: 15.1
J. Zhang et al. , BioRxiv , 10.1101/2021.08.17.456689. ⁴	G614, Gamma, Kappa, and Delta	BLI	AR2G biosensors	Full-length S protein	Dimeric ACE2			G614: 124.0 Alpha: 45.9 Beta: 71.4
					Monomeric ACE2			G614: 19.6 Gamma: 2.4 Kappa: 7.5 Delta: 176
L. Yurkovetskiy et al. , Cell , 10.1016/j.cell.2020.09.032. ⁸	WT and G614	SPR	Series S CM5 sensor chip	ACE2-Fc	Soluble spike trimer protein	25		G614: 288.0 Gamma: 14.8 Kappa: 160.0 Delta: 208.0
						37		WT: 1.38 G614: 7.97 WT: 0.71 G614: 3.76
S. Ozono et al. , Nat Commun , 10.1038/s41467-021-21118-2. ⁹	WT and G614	BLI	AHC biosensor	ACE2-Fc	Stabilized S ectodomain	25		WT: 0.0146 G614: < 0.001
						30		WT: 0.614 G614: 0.297
						37		WT: 1.58 G614: 1.04
S. M.-C. Gobeil et al. , Science , 10.1126/science.abi6226. ¹⁰	G614, Alpha, and Beta	SPR	Series S Strepavidin (SA) chip	Stabilized S ectodomain	ACE2-Fc			G614: 218.29 Alpha: 45.20 Beta: 72.63
M. McCallum et al. , BioRxiv , 10.1101/2021.08.11.455956. ¹¹	Alpha, Kappa, Delta and Delta+	BLI	SA biosensors	His-avi-tagged RBD	Monomeric ACE2			WT: 147 Alpha: 26 Kappa: 88 Delta: 180 Delta+: 550

Q2-5. In particular, it is unclear why the authors used a different concentration range for G614-S: maximum ACE2 concentration of 1800 nM instead of 200 nM as used in the other three variants (Fig. 1A). Such a concentration is unusually high for ACE2 binding analyses compared to the literature. The authors should keep the data presentation consistent unless there is a clear reason to make specific changes to only one of the four S variants.

A2-5: We understand the reviewer's concern. Note that for G614-S, the signal values generated at ACE2 concentrations of 200, 66.7, and 22.2 nM were greater than 0.01, while at ACE2 concentration of 7.41 nM, the signal value was close to 0. Accurate K_D value calculation requires at least 5 concentrations-produced reaction curves. In order to meet this requirement, for the G614-S affinity measurement, we increased the maximum ACE2 concentration to 1800 nM to obtain trustable raw data with good fitting (please see the Fig.1A, also shown below) for K_D calculation. We have now described the reason of making specific changes to G614-S variant into the Method section in our revised manuscript (please see Page 15, Lines 410-413), to read "For WT-S, Kappa-S and Beta-S, ACE2 concentration range used was 200 to 0.823 nM, while for G614-S variant, ACE2 concentration range was 1800 to 7.41 nM, since at ACE2 concentration of 7.41 nM, the signal value was already close to 0."

A

Fig.1 Characterization of properties of the S proteins of Beta and Kappa variants. (A) Measurement of binding affinity between S trimers and ACE2 using bio-layer interferometry (BLI). Association and dissociation steps are divided by dotted lines. Raw sensorgram curves and fitting curves were shown in color and black, respectively.

Q2-6. With regard to the definition of conformational changes in the S protein structures, the authors replied that the command 'measure rotation' in Chimera was used to calculate the rotation angles. The output of the function contains more

informational than the single rotation angle. According to the manual of Chimera, the function will also report the transformation of model2 relative to model1 as:

* a matrix in which the first three columns describe a rotation and the fourth describes a translation (performed after the rotation)

* an axis of rotation (a unit vector), point on the axis, rotation angle, and shift parallel to the axis

The latter parameter, namely the axis of rotation, is important as it will define how the rotation angle is calculated with respect to which axis. Such information is necessary for others to reproduce the calculations in future studies. It is particularly important when the RBD/NTD motions are described by different geometrical definitions, and comparisons between this study and their studies need to be made.

A2-6: We have now followed the suggestion to display the rotation axis generated through the “measure rotation” command in Chimera for RBD/NTD motions in Fig. S5C and Fig. S2G in the revised manuscript. For the convenient of the reviewer and editor, we also show them below as Fig. R1A and R1B, respectively.

Fig. R1 (A) Protomer 1 from the overlaid structures of Kappa S-open (in green) and G614 (PDB: 7KRR, purple), and zoomed-in view of the NTD-SD2 region, with the NTD rotation hinge/axis, located around the region spatially between NTD and SD2, indicated by a blue rod. (B) The 3DVA mode 1 motion of Kappa S-ACE2 displayed in maps, showing two extreme conditions in the motion, with the rotation hinge/axis for RBD, around the lower part of SD1, labeled as a blue rod.

In other studies^{12,13}, centroids of domains were generally used to define vectors, both of which were then employed to measure relative motion of a certain domain between two conformers, in a way somewhat different from ours. For example, in one study¹², the centroids of SD2 and SD1 were used as pivot point to define the rotation of NTD and RBD, respectively, between two conformations (Fig. R2). Their choice of pivot points is close to the position of our rotation axis for NTD and RBD. We have now included this description in Method in our revised manuscript (Line 519-523 on Page18).

Fig. R2 Comparison of the relative domain orientations of RBD-up and RBD-down conformations¹².

Minor comments:

1. The choice of colors in Fig. 2F, Fig. 3D and Fig. 3E should be improved as some of the colors are very similar, making it challenging to visually compare the differences.

Thanks for the suggestion. We have modified the model rendering colors in Fig. 2F, Fig. 3D and Fig. 3E for better visual comparison. For the convenience of the reviewer and editor, we also show it below (from left to right: Fig. 2F, Fig. 3D, and Fig. 3E).

References:

- 1 Yang, T.-J. *et al.* Structure-activity relationships of B.1.617 and other SARS-CoV-2 spike variants. *BioRxiv*, doi:10.1101/2021.09.12.459978 (2021).
- 2 Saville, J. W. *et al.* Structural and Biochemical Rationale for Enhanced Spike Protein Fitness in Delta and Kappa SARS-CoV-2 Variants. *BioRxiv*, doi:10.1101/2021.09.02.458774 (2021).
- 3 Ren, W. *et al.* Characterization of SARS-CoV-2 variants B.1.617.1 (Kappa), B.1.617.2 (Delta) and B.1.618 on cell entry, host range, and sensitivity to convalescent plasma and ACE2 decoy receptor. *BioRxiv*, doi:10.1101/2021.09.03.458829 (2021).
- 4 Zhang, J. *et al.* Membrane fusion and immune evasion by the spike protein of SARS-CoV-2 Delta variant. *bioRxiv*, doi:10.1101/2021.08.17.456689 (2021).
- 5 Cai, Y. *et al.* Structural basis for enhanced infectivity and immune evasion of SARS-CoV-2 variants. *Science*, doi:10.1126/science.abi9745 (2021).
- 6 Zhang, J. *et al.* Structural impact on SARS-CoV-2 spike protein by D614G substitution. *Science* **372**, 525-530, doi:10.1126/science.abf2303 (2021).
- 7 Xu, C. *et al.* Conformational dynamics of SARS-CoV-2 trimeric spike glycoprotein in complex with receptor ACE2 revealed by cryo-EM. *Sci Adv* **7**, doi:10.1126/sciadv.abe5575 (2021).
- 8 Yurkovetskiy, L. *et al.* Structural and Functional Analysis of the D614G SARS-CoV-2 Spike Protein Variant. *Cell* **183**, 739-751 e738, doi:10.1016/j.cell.2020.09.032 (2020).
- 9 Ozono, S. *et al.* SARS-CoV-2 D614G spike mutation increases entry

- efficiency with enhanced ACE2-binding affinity. *Nat Commun* **12**, 848, doi:10.1038/s41467-021-21118-2 (2021).
- 10 Gobeil, S. M. *et al.* Effect of natural mutations of SARS-CoV-2 on spike structure, conformation, and antigenicity. *Science*, doi:10.1126/science.abi6226 (2021).
- 11 McCallum, M. *et al.* Molecular basis of immune evasion by the delta and kappa SARS-CoV-2 variants. *bioRxiv*, doi:10.1101/2021.08.11.455956 (2021).
- 12 Yang, T. J. *et al.* Effect of SARS-CoV-2 B.1.1.7 mutations on spike protein structure and function. *Nat Struct Mol Biol* **28**, 731-739, doi:10.1038/s41594-021-00652-z (2021).
- 13 Gobeil, S. M. *et al.* Effect of natural mutations of SARS-CoV-2 on spike structure, conformation, and antigenicity. *Science* **373**, doi:10.1126/science.abi6226 (2021).